# Visually induced changes in cytokine production in the chick choroid

**Jody A Summers\*, Elizabeth Martinez**

Department of Cell Biology, University of Oklahoma Health Sciences Center, Oklahoma City, United States

**Abstract** Postnatal ocular growth is regulated by a vision-dependent mechanism that acts to minimize refractive error through coordinated growth of the ocular tissues. Of great interest is the identification of the chemical signals that control visually guided ocular growth. Here, we provide evidence that the pro-inflammatory cytokine, interleukin-6 (IL-6), may play a pivotal role in the control of ocular growth using a chicken model of myopia. Microarray, real-time RT-qPCR, and ELISA analyses identified IL-6 upregulation in the choroids of chick eyes under two visual conditions that introduce myopic defocus and slow the rate of ocular elongation (recovery from induced myopia and compensation for positive lenses). Intraocular administration of atropine, an agent known to slow ocular elongation, also resulted in an increase in choroidal IL-6 gene expression. Nitric oxide appears to directly or indirectly upregulate choroidal IL-6 gene expression, as administration of the non-specific nitric oxide synthase inhibitor, L-NAME, inhibited choroidal IL-6 gene expression, and application of a nitric oxide donor stimulated IL-6 gene and protein expression in isolated chick choroids. Considering the pleiotropic nature of IL-6 and its involvement in many biological processes, these results suggest that IL-6 may mediate many aspects of the choroidal response in the control of ocular growth.

**\*For correspondence:**
jody-summers@ouhsc.edu

**Competing interest:** The authors declare that no competing interests exist.

## Introduction

High myopia is a significant risk factor for several blinding eye diseases including glaucoma, retinal detachment, and macular degeneration, and therefore represents a leading cause of blindness worldwide (*Buch et al., 2001*). The prevalence of myopia is continuing to increase and is predicted to affect nearly half of the global population by 2050 (*Holden et al., 2016*). Although clinical and experimental studies indicate that normal eye growth (emmetropization) is controlled by visual input (*Wallman and Winawer, 2004*), the molecular mechanisms underlying myopia development in humans is not understood.

Animal models have provided valuable insights into the role of the visual environment in ocular growth control. Deprivation of form vision, through the use of visual 'occluders' or 'goggles' results in accelerated ocular growth and the development of myopia within a matter of days in many vertebrate species, including fish, chicks, mice, tree shrews, guinea pigs, and primates (*Howlett and McFadden, 2006*; *Shen et al., 2005*; *Schaeffel et al., 2004*; *Tejedor and de la Villa, 2003*; *Troilo and Judge, 1993*; *Wallman et al., 1978*; *Sherman et al., 1977*). Upon removal of the occluder, the elongated eye now experiences myopic defocus, which results in a rapid deceleration in ocular elongation and eventual return to emmetropia (recovery) (*Wallman and Adams, 1987*).

Postnatal ocular growth can also be manipulated through the application of positive or negative lenses, as the eye has been shown to compensate for the imposed defocus in many vertebrates, including fish, chicks, mice, mammals, and primates (*Tkatchenko et al., 2010*; *Shen and Sivak, 2007*; *Norton et al., 2006*; *Graham and Judge, 1999*; *Hung et al., 1995*; *Schaeffel et al., 1988*). Application of positive lenses results in images forming in front of the retina (myopic defocus). This imposed myopic defocus causes slowing of the rate of ocular elongation and thickening of the choroid, which

effectively pushes the retina toward the image plane, thereby minimizing the imposed refractive error (*Wallman et al., 1995*). Conversely, application of negative lenses moves the image plane behind the retina (hyperopic defocus) and results in an increased rate of axial elongation and thinning of the choroid to pull the retina back toward the image plane.

It is well established that visually induced changes in ocular growth are the result of a locally driven 'retina-to-choroid-to-scleral molecular signaling cascade' that is initiated by a visual stimulus, followed by biochemical and structural changes in the retina and choroid, ultimately resulting in altered extra-cellular matrix (ECM) remodeling of the scleral shell (*Troilo and Smith, 2019*; *Tkatchenko et al., 2006*; *Fischer and Reh, 2000*). The choroid, a highly vascularized layer located immediately adjacent to the sclera, has been shown to undergo changes in thickness, permeability, and blood flow during periods of visually guided eye growth (*Rada and Palmer, 2007*; *Fitzgerald et al., 2002*; *Pendrak et al., 2000*; *Wallman et al., 1995*). Moreover, due to its proximity to the sclera, the choroid is suspected to synthesize and/or release scleral growth regulators to control the rate of ocular elongation in response to visual stimuli (*Rada and Palmer, 2007*; *Marzani and Wallman, 1997*). All-*trans*-retinoic acid is one potential choroidally derived scleral growth regulator, whose choroidal concentrations are modulated by the activity of retinaldehyde dehydrogenase 2 (RALDH2) (*Rada et al., 2012*; *Mertz and Wallman, 2000*). Of much interest, therefore, is the identification of genes causally involved in the regulation of the choroidal response during visually guided eye growth.

Here, we report rapid and significant changes in choroidal gene expression of the cytokine, inter-leukin-6 (IL-6), in response to myopic defocus and in response to chemical compounds known to modulate eye growth. Considering the pleiotropic nature of IL-6 and its involvement in many biological processes, these results suggest that IL-6 may mediate many aspects of the choroidal response in the control of ocular growth.

## Results

### Expression of IL-6 in chick ocular tissues

Immunohistochemical staining for IL-6 indicated that IL-6 is expressed in numerous cells throughout the choroid and RPE as punctate cytoplasmic deposits, some of which appeared to colocalize with nuclei of RPE and choroidal cells (*Figure 1A and B*, and *Figure 1—figure supplement 1*, *Figure 1—figure*

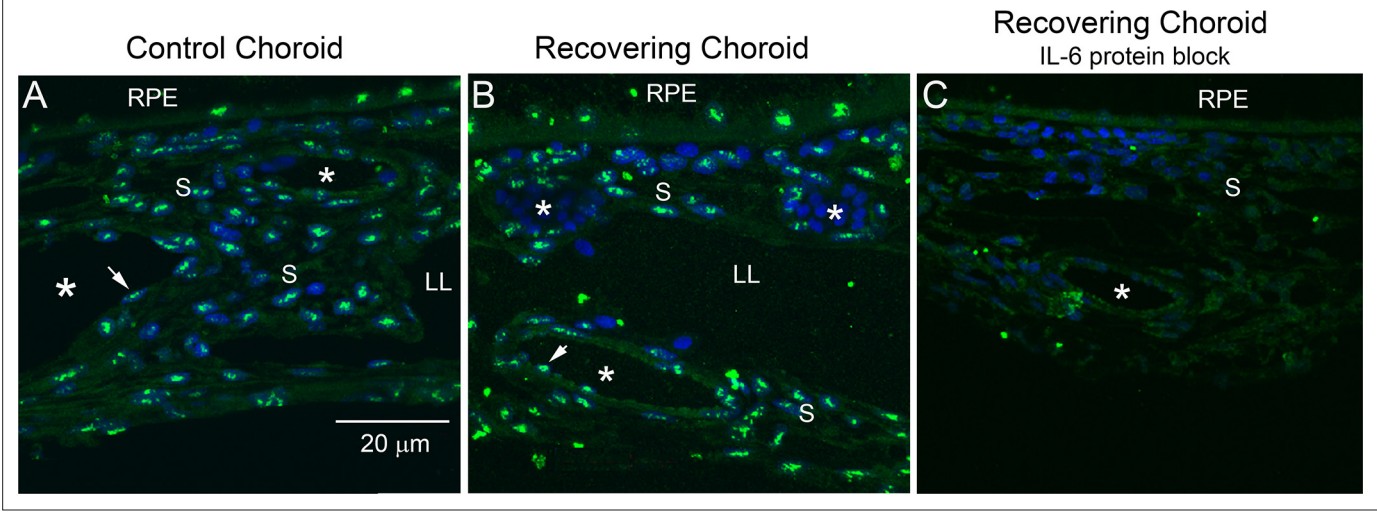

**Figure 1.** Immunohistochemical localization of IL-6 in chick choroids. (**A, B**) Il-6 was localized in treated and contralateral control eyes after 24 hr of recovery from induced myopia (green labeling). (**C**) Preabsorption of anti-IL-6 with a tenfold molar excess of recombinant chicken IL-6 (1.67 µM) before use on tissue sections abolished IL-6 labeling. Bar=20 µm in (**A–C**). Choroidal blood vessels are indicated by asterisks (*). Vascular endothelium is indicated by arrowheads (↑). IL-6, interleukin-6 ; LL, lymphatic lacunae; RPE, retinal pigmented epithelium; S, extravascular choroidal stroma.

The online version of this article includes the following figure supplement(s) for figure 1:

**Figure supplement 1.** Immunolocalization of IL-6 in the chick choroid.

**Figure supplement 2.** Improved visualization of DAPI labeling in chick RPE and choroid.

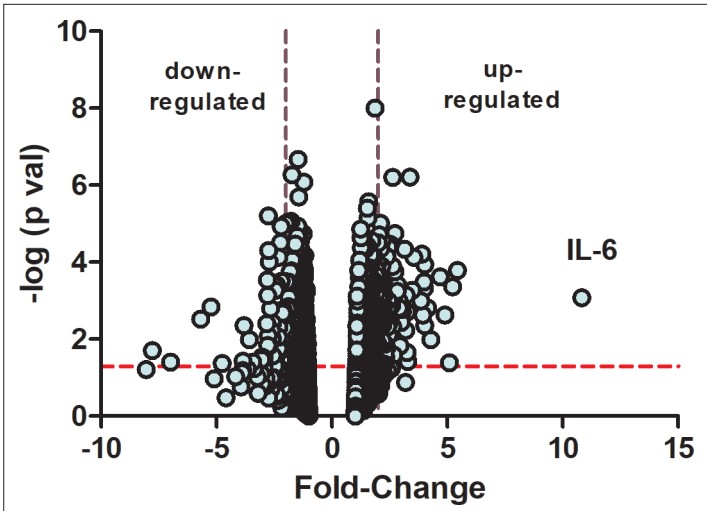

**Figure 2.** Microarray identifies IL-6 as a gene highly overexpressed in early recovery. A volcano plot of Affymetrix chicken microarray data indicated that 207 genes were found to be significantly differentially expressed by ≥2-fold in recovering choroids as compared with choroids from normal untreated chicks (p≤0.05). The horizontal dashed red line indicates where p=0.05, with points above the line having p<0.05 and points below the line having p>0.05. The area between the dashed purple lines indicates points having a fold-change less than |2|. IL-6 was increased by 10.83-fold in recovering choroids compared with normal choroids (n=5 birds in each group); p=0.00084, one-way ANOVA using Method of Moments. IL-6, interleukin-6.

The online version of this article includes the following source data and figure supplement(s) for figure 2:

**Source data 1.** Microarray data.

**Figure supplement 1.** Refractions of form-deprived (FD; right eyes) and untreated contralateral control eyes (left eyes) following 10 days of form deprivation.

**Figure supplement 1—source data 1.** Refractions of form-deprived and untreated contralateral control eyes.

---

*supplement 2*). IL-6-containing cells included the RPE, choroidal endothelial cells (arrowheads; blood vessels are labeled with asterisks), and choroidal stromal cells. Immunolabeling was abolished after preabsorption of this antibody with a tenfold molar excess of chicken IL-6 demonstrating that the immunohistochemical detection procedure was specific (*Figure 1C*).

## IL-6 is upregulated in response to myopic defocus

### 1. Form deprivation and recovery

Following 10 days of form deprivation, chick eyes became elongated and developed significant myopia (see *Figure 2—figure supplement 1*). In response to the myopic defocus, the choroid undergoes a number of structural and chemical changes that result in recovery from the imposed myopia (*Wallman et al., 1995*). Results from an Affymetrix microarray experiment indicated that IL-6 was increased over tenfold in choroids of chick eyes following 6 hr of recovery, compared with normal, untreated eyes (*Figure 2*).

To determine the precise temporal pattern of IL-6 expression during recovery, we utilized TaqMan real-time PCR to quantify IL-6 mRNA concentrations in choroids following 10 days of form deprivation and over several time points during recovery from induced myopia (*Figure 3A*). IL-6 mRNA was significantly increased in choroids following 90 min to 24 hr of recovery compared to contralateral control eyes, reaching a maximum following 6 hr of recovery. By 4 days of recovery, IL-6 mRNA was significantly downregulated in treated choroids, compared with that of treated choroids at 24 hr of recovery, and was similar to that of fellow control eyes p=0.19 and 0.08, for 4 and 8 days of recovery, respectively, (Wilcoxon signed-rank test for matched pairs).

The rapid increase in choroidal IL-6 gene expression observed during recovery prompted us to determine whether increased choroidal IL-6 gene expression was an artifact of removal of the occluder, rather than due to a visual stimulus. To address this possibility, one group of chicks was kept in complete darkness for 6 hr following the removal of the occluder (*Figure 3A*, 6 hr in dark).

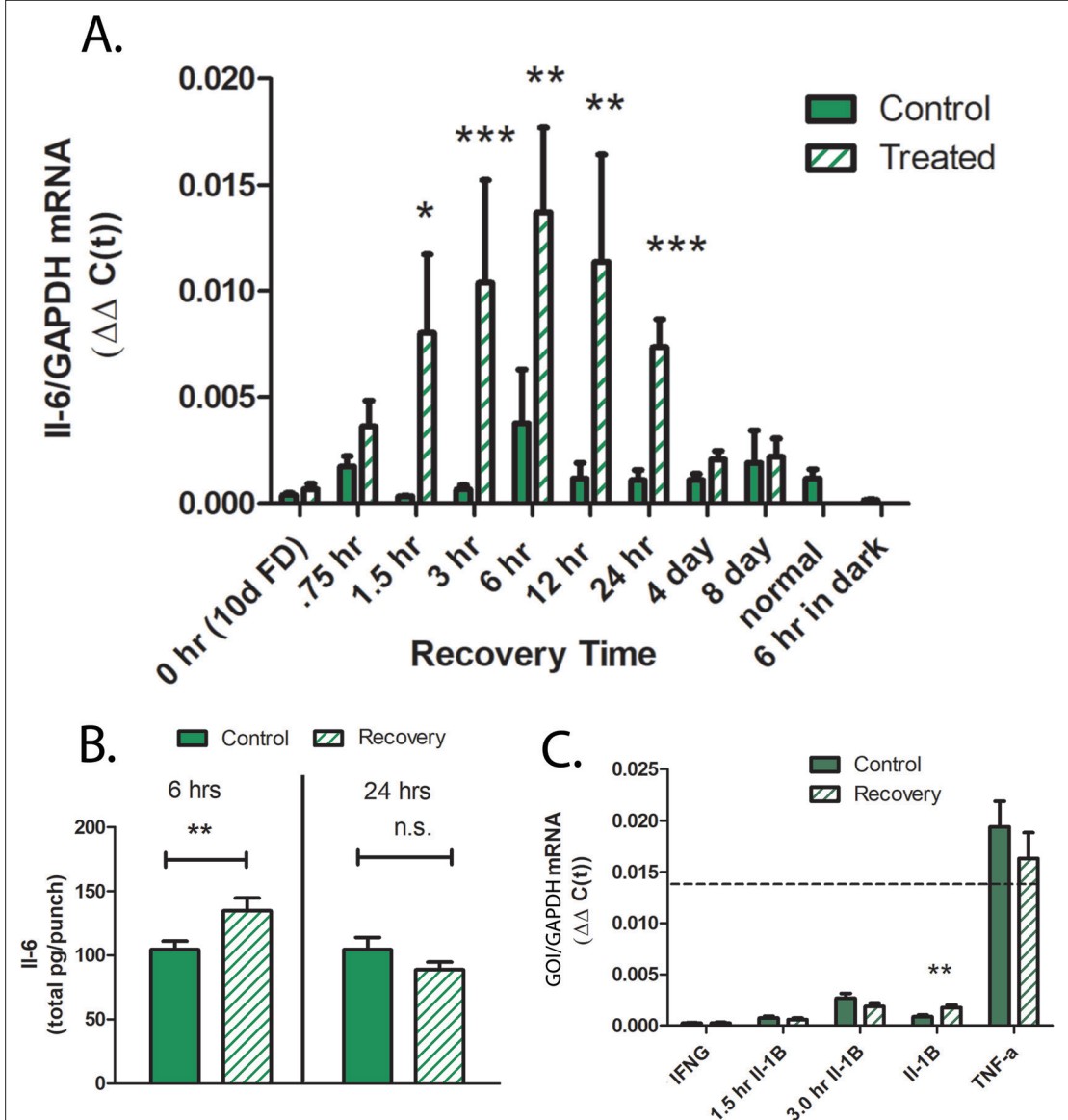

**Figure 3.** Cytokine gene and protein expression in chick choroids. (**A**) IL-6 mRNA expression in choroids from control and treated eyes, following 10 days of form deprivation (0 hr/10 days FD), 0.75 hr to 8 days of recovery from form deprivation, normal, untreated eyes (normal), and in eyes recovered for 6 hr, but kept in total darkness (6 hr in dark) (n=5–16 birds in each group) ***p<0.001, **p<0.01, *p=0.013, Wilcoxon signed-rank test for matched pairs. (**B**) IL-6 protein production by control and recovering choroids following 6 and 24 hr of recovery from induced myopia. Data are expressed as mean ± SEM (n=16) **p=0.0102, paired t-test. (**C**) Quantification of other proinflammatory cytokines in chick choroids. Gene expression of Interferon gamma (IFNG), interleukin-1B (IL-1B), and tumor necrosis factor alpha (TNF-α) was quantified in control and treated chick choroids following 6 hr of recovery. Additionally, IL-1B mRNA was quantified following 1.5 and 3 hr of recovery. The dashed line indicates the average IL-6 expression in 6 hr recovering choroids (n=6–11 birds in each group) **p=0.0059, Wilcoxon signed-rank test for matched pairs. IL-6, interleukin-6.

The online version of this article includes the following source data and figure supplement(s) for figure 3:

**Source data 1.** Cytokine gene and protein expression in chick choroids.

**Figure supplement 1.** IL-6 has no direct effect on scleral proteoglycan synthesis.

**Figure supplement 1—source data 1.** IL-6 on scleral proteoglycan synthesis.

Interestingly, IL-6 gene expression was significantly lower in both control and recovering eyes, as compared with IL-6 mRNA levels from all other choroids of control, recovering or form deprived eyes reared under normal room light (p=0.013, Mann-Whitney test, for choroids of dark reared control or recovering eyes compared with the lowest control group [1.5 hr control group]).

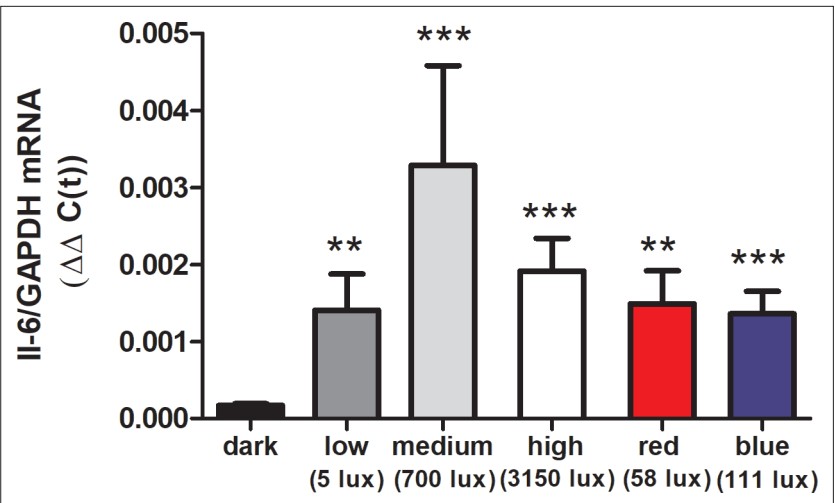

**Figure 4.** Effect of light intensity on IL-6 mRNA expression. Normal chicks were housed in complete darkness (dark), white LED light of varying intensities ('low,' 5 lux; 'medium,' 700 lux; 'high,' 3150 lux), red LED light ('red,' 58 lux), or blue LED light ('blue,' 111 lux) for 6 hr at which time choroids were isolated with Il-6 mRNA was quantified by TaqMan real-time PCR (n=6–8 birds [12–16 choroids]) in each group. ***p<0.001, **p<0.01, Kruskal-Wallis test with Dunn's multiple comparisons. IL-6, interleukin-6.

The online version of this article includes the following source data and figure supplement(s) for figure 4:

**Source data 1.** Results of Taqman (IL-6/GAPDH ΔΔC(t)) Light Intensity Experiment.

**Figure supplement 1.** Spectra of LED light sources as a function of wavelength.

**Figure supplement 1—source data 1.** Spectra of LED light sources as a function of wavelength.

Choroidal protein expression of IL-6 was also significantly increased following 6 hr of recovery, compared to contralateral control eyes (↑1.36-fold ±0.47 sd, p=0.0102, paired t-test), but returned to control levels by 24 hr of recovery (*Figure 3B*). We also evaluated gene expression of the chicken cytokines, interferon gamma (IFN-γ), interleukin-1β (IL-1β), and tumor necrosis factor alpha (TNF-α) in choroids of eyes following 1.5–6 hr of recovery and in contralateral control eyes. Gene expression of TNF-α was substantially higher (≈7-fold) than all other cytokines examined, but not significantly different between control and recovering eyes. Only gene expression of IL-1β following 6 hr of recovery was significantly elevated in recovering eyes compared with controls (↑2.79-fold ±2.44 sd, p=0.0059, Wilcoxon signed-rank test for matched pairs) (*Figure 3C*).

## 2. Light intensity

Based on our observation that IL-6 mRNA was significantly lower in choroids of birds kept in darkness for 6 hr, compared to control or treated eyes reared under standard room lighting, we evaluated the effect of varied light intensity on choroidal IL-6 gene expression. Normal untreated chicks were kept in dim light (5 lux), medium intensity light (700 lux), and high intensity light (3150 lux), as well as red LED light (58 lux) and blue LED light (111 lux) for 6 hr prior to RNA isolation. Exposure to all light intensities resulted in a significant increase in IL-6 mRNA, compared to IL-6 gene expression in choroids of dark reared chicks (*Figure 4*); however, no differences were observed in IL-6 mRNA levels between the five lighting conditions, with all IL-6 mRNA values similar to that of the normal untreated chick choroids (*Figure 3A*).

## 3. Optical defocus

Following removal of the occluder, previously form deprived eyes experience myopic defocus due to form deprivation-induced myopia. We therefore determined whether choroidal IL-6 gene expression was affected following a period of imposed myopic or hyperopic defocus via the application of +15 D or –15 D spectacle lenses (*Figure 5A and B*). Following 24 hr of +15 D lens wear, choroidal IL-6 gene expression was significantly increased compared with contralateral control eyes (↑8.5-fold ±17.18 sd, p=0.0003, Wilcoxon signed-rank test for matched pairs) (*Figure 5C*). No significant differences were

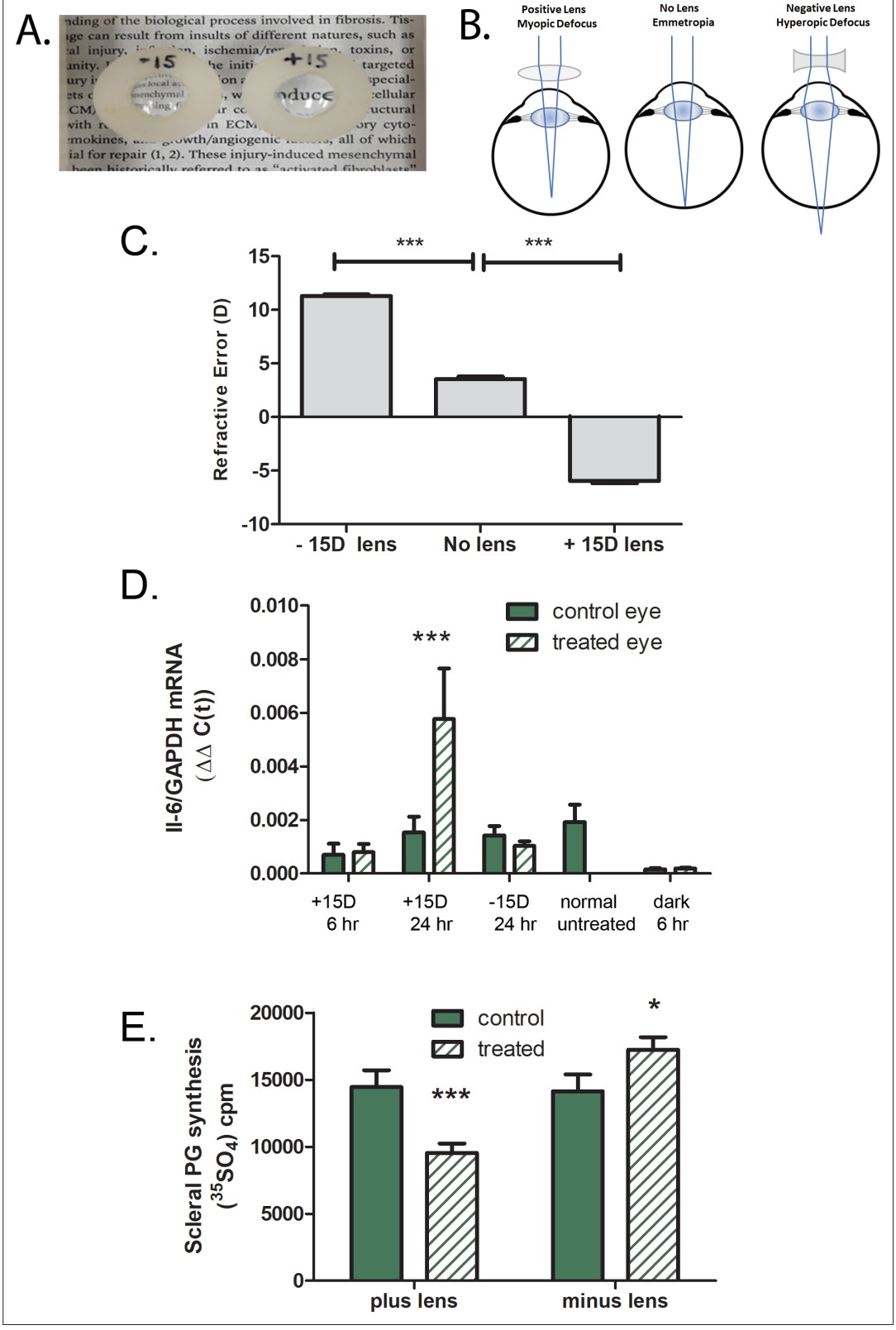

**Figure 5.** Effect of imposed defocus on choroidal IL-6 gene expression. (**A**). Spectacle lenses [minus 15 D (–15) or plus 15 D (+15)] were applied to the right eyes of chicks for 6–24 hr. (**B**). Schematic diagram illustrating the effects of imposed optical defocus on the location of ocular images of distant objects for an emmetropic eye (center); positive lenses move the image plane in front the retina, imposing myopic defocus (left), while negative lenses

*Figure 5 continued on next page*

*Figure 5 continued*

move the image plane behind the retina, imposing hyperopic defocus (right). (**C**) Refractive status of chick eyes while wearing –15 D and +15 D lenses. Application of –15 D lenses results in a hyperopic shift in the refraction of normal chick eyes, relative to untreated (no lens) eyes, whereas application of +15 D lenses results in a myopic shift in refraction, compared to untreated eyes. ***p<0.0001, ANOVA with Bonferroni correction for n=2 chicks in each group (five measurements/chick). (**D**) IL-6 mRNA expression in choroids from control and treated eyes, following 6 or 24 hr of plus lens wear (n=6 and n=27, respectively), 24 hr of minus lens wear (n=34), and normal untreated choroids (n=8). ***p=0.0003, Wilcoxon signed-rank test for matched pairs. (**E**) Scleral proteoglycan synthesis following 24 hr of lens wear. Proteoglycan synthesis was significantly reduced following 24 hr of + 15 D lens wear, compared to untreated contralateral control eyes (***p=0.00047, paired t-test, n=10) and was significantly increased following 24 hr of –15 D lens wear, compared with untreated contralateral control eyes (*p=0.0403, paired t-test, n=13).

The online version of this article includes the following source data for figure 5:

**Source data 1.** Effect of imposed defocus on choroidal IL-6 gene expression.

---

detected in IL-6 gene expression following 6 hr of +15 D lens wear. Treatment with –15 D lenses had no statistically significant effect on choroidal IL-6 gene expression, although a trend toward decreased expression was noted. Scleral proteoglycan synthesis was also assessed following 24 hr of lens treatment to confirm that the +15 D and –15 D lenses were inducing compensatory ocular growth responses (*Figure 5D*). As expected, treatment with +15 D lenses resulted in a significant decrease in scleral proteoglycan synthesis (p=0.00047, paired t-test) and treatment with –15 D lenses resulted in a significant increase in scleral proteoglycan synthesis (p=0.0403, paired t-test).

## Choroidal IL-6 mRNA expression in response to nitric oxide

*Nickla et al., 2009* have previously demonstrated that nitric oxide synthesis is necessary for compensation for imposed myopic defocus. Administration of the non-specific inhibitor NOS inhibitor, Nᵃ-nitro-L-arginine methyl ester (L-NAME), or the nNOS inhibitor Nʷ -propyl-L-arginine, blocks recovery from form deprivation myopia (FDM), or compensation to +10 D lens-induced defocus due to inhibition of choroidal thickening and dis-inhibition of scleral proteoglycan synthesis (*Nickla et al., 2009*; *Nickla and Wildsoet, 2004*). We therefore investigated the role of nitric oxide on choroidal IL-6 transcription using several approaches. First, L-NAME, or vehicle, was administered via intravitreal injection to chick eyes following 10 days of form deprivation. Chicks were then given unrestricted vision for 6 hr and choroidal IL-6 mRNA was quantified (*Figure 6A*). Following 6 hr of recovery, IL-6 mRNA

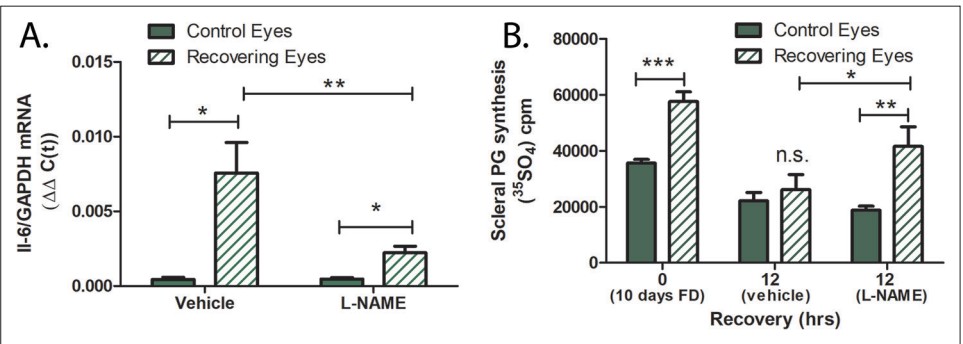

**Figure 6.** L-NAME inhibits choroidal IL-6 transcription and recovery. (**A**) Intravitreal injection of L-NAME (16.2 µmol/eye) immediately prior to recovery significantly reduced IL-6 mRNA levels compared to recovering eyes receiving vehicle only (0.9% NaCl) (**p=0.007, Mann-Whitney U-test, n=7; *p=0.015, Wilcoxon signed-rank test for matched pairs, n=7). (**B**) L-NAME disinhibits scleral proteoglycan synthesis in recovering eyes. Following 12 hr of recovery from 10 days of form deprivation (FD), scleral proteoglycan synthesis decreased to control levels in vehicle-treated eyes, but remains significantly increased over control levels in L-NAME treated eyes (***p<0.0001, paired t-test, n=16; **p=0.0121 Wilcoxon signed-rank test for matched pairs, n=17; *p=0.0421, Mann-Whitney U-test, n=17). IL-6, interleukin-6.

The online version of this article includes the following source data for figure 6:

**Source data 1.** The effect of L-NAME on choroidal IL-6 transcription and recovery.

was significantly increased in recovering eyes of vehicle (saline) treated eyes compared with contralateral control eyes (↑12-fold ±5.23 sd, p=0.015, Wilcoxon signed-rank test for matched pairs). Administration of L-NAME just prior to recovery resulted in a significant decrease in choroidal IL-6 mRNA, 6 hr following L-NAME administration, as compared with choroidal IL-6 mRNA levels in recovering eyes of saline-treated eyes (p=0.007, Mann-Whitney U-test). L-NAME administration did not completely abolish the recovery-induced rise in choroidal IL-6 mRNA; IL-6 mRNA levels in choroids of L-NAME-treated eyes were significantly higher than that of contralateral untreated eyes (p=0.015, Wilcoxon signed-rank test for matched pairs). As previously reported (*Summers Rada and Hollaway, 2011*), scleral proteoglycan synthesis was significantly increased in the posterior sclera of chick eyes during the development of FDM (Day 0 of recovery) (p<0.0001, paired t-test) and was rapidly downregulated following 12 hr of recovery to levels similar to that of contralateral control eyes (vehicle) (*Figure 6B*). Intravitreal application of L-NAME inhibited this recovery response, resulting in a significant increase in scleral proteoglycan synthesis in recovering eyes, as compared with contralateral control eyes (p=0.0121, Wilcoxon signed-rank test for matched pairs) and compared with recovering eyes of vehicle-treated chicks (p=0.0421, Mann-Whitney U-test). These results confirm that intravitreal administration of L-NAME in our study resulted in the same effects on eye growth as have been previously reported.

L-NAME administration attenuated the recovery-induced increase in choroidal IL-6 transcription observed following 6 hr of recovery, suggesting that NO is involved in the regulation of choroidal IL-6 mRNA transcription. Therefore, we directly tested the effect of an NO donor on IL-6 gene transcription using isolated chicken choroids (*Figure 7*). Treatment of choroids with PAPA-NONOate, an NO donor with a half-life of 15 min at 37°C, led to a concentration dependent increase in IL-6 mRNA that reached a five-fold increase at 1.5 mM (*Figure 7A*). Protein expression of IL-6 was also significantly increased in isolated choroids following incubation with PAPA-NONOate, compared to that of choroids incubated in culture medium alone (p=0.0079, p=0.0.0357, for control vs. 3 mM PAPA-NONOate and control vs. 5 mM PAPA-NONOate, respectively; Mann-Whitney U-test) (*Figure 7B*).

As NO has been shown to activate members of the MAPK pathway in a cGMP-independent

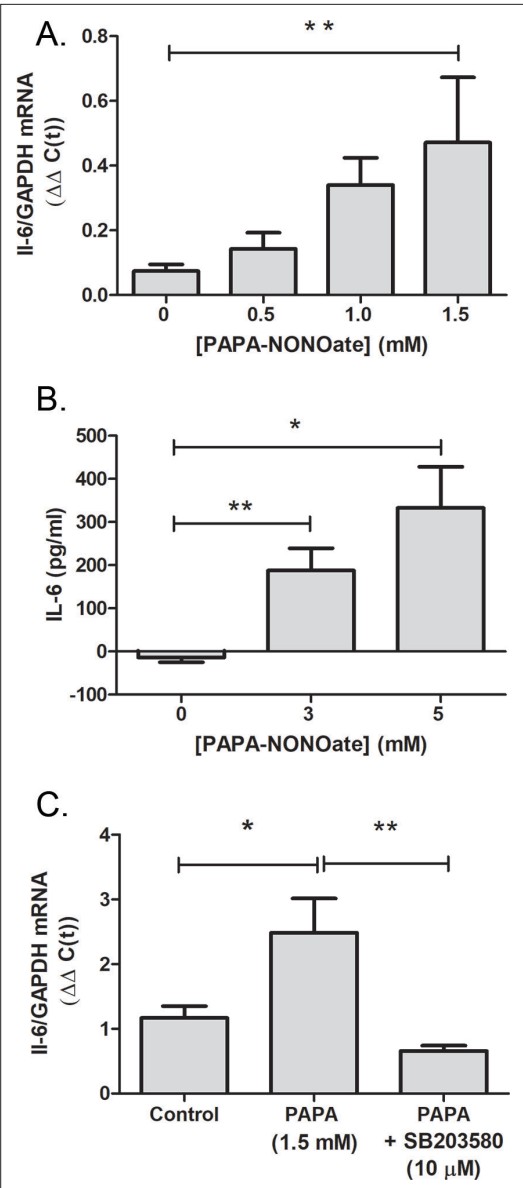

**Figure 7.** The NO donor, PAPA-NONOate, stimulates choroidal IL-6 production. Choroids were isolated from normal chicken eyes and were incubated with the indicated concentrations of PAPA-NONOate for 24 hr. (**A**) IL-6 gene expression was significantly increased in choroids following incubation in 1.5 mM PAPA-NONOate (*p=0.0079, Mann-Whitney U-test, n=4–5 choroids in each group). (**B**) IL-6 protein concentrations were significantly increased in choroid culture supernatants following incubation with 3–5 mM PAPA-NONOate (**p=0.0079, *p=0.0357, Mann-Whitney U-test, n=3–5 choroids in each group). (**C**) Incubation of chicken choroids with PAPA-NONOate (1.5 mM) together with the p38 MAPK inhibitor SB203580 (10 µM) abolished the PAPA-NONOate-induced increase in IL-6 mRNA (**p=0.0032, *p=0.0308, Student's t-test, n=10 choroids in each group). IL-6, interleukin-6.

The online version of this article includes the following

*Figure 7 continued*

source data for figure 7:

**Source data 1.** The effect of the NO donor, PAPA-NONOate on choroidal IL-6 production.

manner, and given that the p38 pathway plays essential roles in the production of IL-6 and other proinflammatory cytokines (IL-1β, TNF-α, and IL-6) (*Guan et al., 1998*), we sought to determine whether p38 MAPK activation contributes to the NO-mediated stimulation of choroidal IL-6 transcription. Treatment of isolated choroids with the p38 specific inhibitor, SB203580, and PAPA-NONOate abolished the NO-induced increase in IL-6 mRNA, suggesting that the NO-stimulated IL-6 transcription is mediated through activation of MAPK signaling pathways (*Figure 7C*).

Since choroidal IL-6 synthesis was upregulated following treatment of choroids with NO donor, PAPA-NONOate, experiments were undertaken to determine if endogenous synthesis of choroidal NO could also stimulate choroidal Il-6 synthesis. Therefore, cultures of isolated chick choroids were incubated with L-arginine, the natural substrate for NO synthesis by the NOS enzymes. Incubation of isolated choroids with L-arginine had no significant effect on choroidal IL-6 gene expression (*Figure 8A*). However, when cultures were also incubated with KCl (50 mM) to depolarize plasma membranes, L-arginine treatment did result in a significant increase in IL-6 mRNA (*Figure 8A and B*).

## Atropine stimulates choroidal IL-6 transcription

Atropine has been shown to be clinically effective at reducing myopia progression in clinical trials (*Upadhyay and Beuerman, 2020*) and in avian and mammalian animal models of myopia, although the mechanism of action is poorly understood (*Whatham et al., 2019*; *McBrien et al., 1993*). Therefore, we examined the effect of atropine on choroidal IL-6 transcription in chicks undergoing FDM (*Figure 9*). Intravitreally delivered atropine (240 nmol/eye) significantly increased choroidal IL-6 mRNA in form deprived eyes as compared with vehicle-treated form deprived eyes (↑4.16-fold ±5.66 sd, p=0.0498, Mann-Whitney U-test), when measured 6 hr following atropine administration (*Figure 9A*). Interestingly, application of atropine (0.1%) directly to isolated chick choroids stimulated IL-6 gene expression (↑4.75-fold ±5.80 sd, p=0.0092, Student's t-test) (*Figure 9B*). Protein expression of IL-6 was also significantly increased in isolated choroids following incubation with 0.1% atropine, compared to that of choroids incubated in culture medium alone (p=0.0491; Mann-Whitney U-test) (*Figure 9C*).

## Discussion

These studies document, for the first time, that the potent inflammatory cytokine IL-6 is expressed in and released by the choroid during the recovery from FDM. Choroidal upregulation of IL-6 is rapid and transient; significant increases in gene expression were observed after only 90 min of unrestricted vision. IL-6 gene expression reaches a maximum level of expression following 6 hr of recovery, and then begins to decline, returning to control levels by 4 days of recovery. The visual stimulus for choroidal IL-6 gene expression is myopic defocus, as a similar rise in IL-6 is observed after 6 hr of positive (+15 D) lens wear. Finally, nitric oxide appears to directly or indirectly upregulate choroidal IL-6 gene expression. We and others have previously shown that changes in the visual environment can cause rapid changes in ocular growth and refraction as evidenced by changes in scleral proteoglycan synthesis, choroidal retinoic acid synthesis, and choroidal thickness (reviewed in *Troilo and Smith, 2019*). The chemical mediators that translate visual signals to scleral ECM remodeling to effect changes in eye size have been only partly identified. IL-6 is a multifunctional cytokine synthesized by a variety of cell types that plays key roles in immune responses, inflammatory reactions, as well as the growth and differentiation of many cell types (*Kishimoto, 2006*). For these reasons, choroidally derived IL-6 could play an important role in the retina-to-sclera signaling cascade. Here, we demonstrate that myopic defocus, as a result of prior form vision deprivation, or due to application of +15 D lenses, stimulates choroidal IL-6 gene expression. In contrast, induction of hyperopic defocus through the application of –15 D lenses resulted in a slight decrease in choroidal IL-6 gene expression, that did not reach statistical significance in the present study.

The observed increase in choroidal IL-6 mRNA was transient; choroidal IL-6 mRNA concentrations peaked at 6 hr following removal of the occluder (myopic defocus), and returned to control levels by 4 days. This transient increase in choroidal IL-6 mRNA is similar to that observed for plasma IL-6

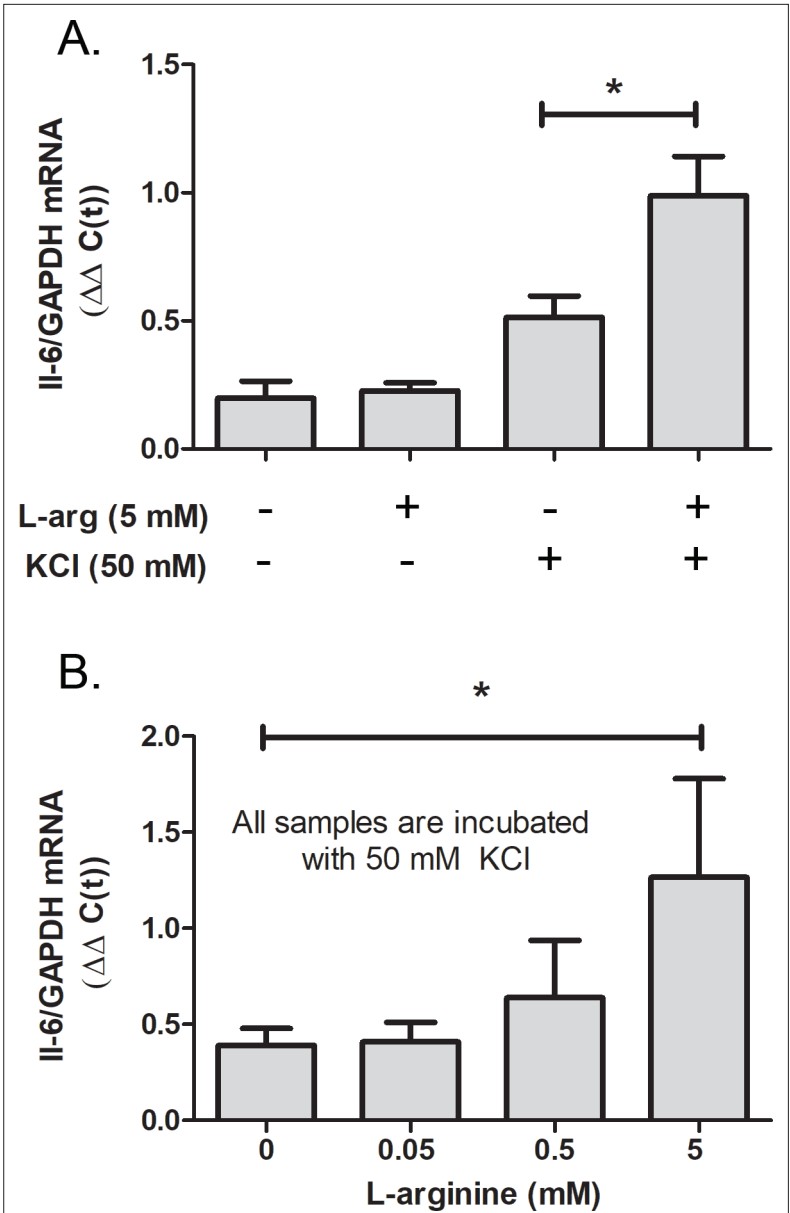

**Figure 8.** L-arginine (L-arg), the NOS substrate, stimulates choroidal IL-6 trasncription. Choroids were isolated from normal chicken eyes and were incubated with the indicated concentrations of L-arg for 24 hr. KCl (50 mM) was added to some cultures to depolarize cell membranes. (**A**) IL-6 gene expression was significantly increased in choroids following incubation in 5 mM L-arg in the presence of 50 mM KCl (*p=0.0188, Mann-Whitney test, n=9 choroids in each group). (**B**) Dose response for the effect of L-arg on IL-6 gene expression. IL-6 gene expression was significantly increased in choroids following incubation in 5 mM L-arginine in the presence of 50 mM KCl (*p=0.04, Mann-Whitney test, n=9 choroids in each group). IL-6, interleukin-6.

The online version of this article includes the following source data for figure 8:

**Source data 1.** The effect of L-arginine on choroidal IL-6 transcription.

following strenuous exercise (*Ostrowski et al., 1998*), following ischemic brain injuries (*Hagberg et al., 1996*; *Fassbender et al., 1994*), surgical trauma (*Nishimoto et al., 1989*), and in a subset of patients with multiple sclerosis following treatment with interferon-β (*Nakatsuji et al., 2006*). It is thought that IL-6 synthesis and release from contracting muscle cells, mononuclear phagocytes, and/ or other cell types initiates a cytokine cascade, inducing the expression of other cytokines, such as IL-1 receptor antagonist, to provide a negative feedback loop to reduce the inflammatory response (*Jordan*

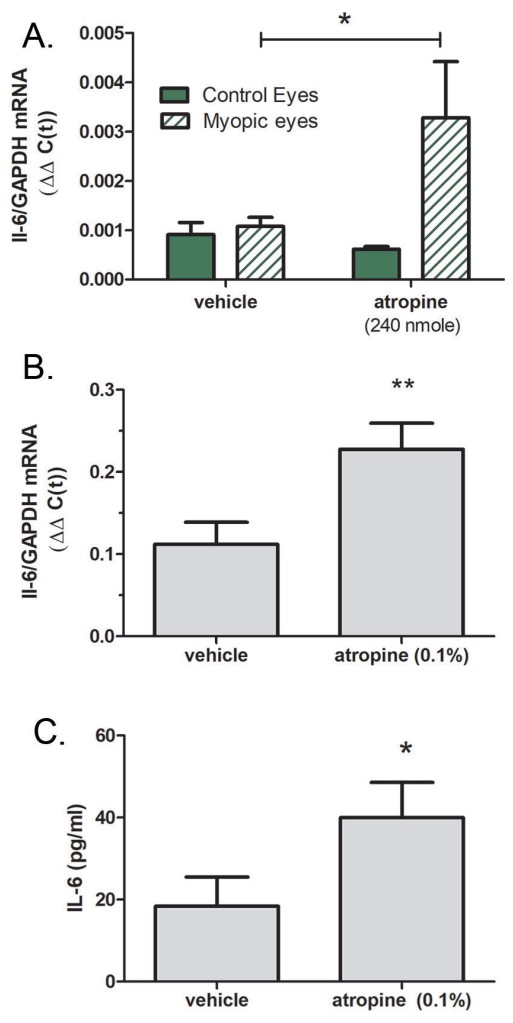

**Figure 9.** Atropine stimulates choroidal IL-6 gene expression. (**A**) Intravitreal injection of atropine (240 nmol/eye) into chick eyes following 14 days of form deprivation (myopic eyes) increased IL-6 mRNA levels compared to myopic eyes receiving vehicle only (PBS) (*p=0.0498, Mann-Whitney U-test, n=18). (**B**) Incubation of chicken choroids in organ culture with 0.15 atropine for 24 hr significantly increased choroidal IL-6 gene expression (**p=0.0092, Student's t-test, n=16). (**C**) IL-6 protein concentrations were significantly increased in choroid culture supernatants following incubation with 0.1% atropine (*p=0.0491, Mann-Whitney U-test, n=10 choroids in each group). IL-6, interleukin-6.

The online version of this article includes the following source data for figure 9:

**Source data 1.** The effect of atropine on choroidal IL-6 gene expression.

et al., 1995). It is unclear as to whether the 6 hr peak in choroidal IL-6 observed in the present study is related to the duration of myopic defocus needed to initiate the maximal IL-6 response, or due to the minimum time required for choroidal IL-6 gene transcription to be upregulated in response to the initial visual stimulus. Our finding of elevated choroidal IL-6 mRNA in treated eyes following only 45 min of myopic defocus together with the previous observation that compensation for positive lenses can occur with as little as 10 min of myopic defocus (*Zhu et al., 2005*), lead us to favor the latter possibility—that is, a very brief period of myopic defocus would likely stimulate a transient increase in choroidal IL-6 transcription, that reaches a peak 6 hr later.

Choroidal IL-6 gene expression was not significantly affected by rearing chicks in varying light intensities (5–3150 lux), or rearing for 6 hrs in red or blue LED light (58 and 111 lux, respectively), but was significantly lower in choroids from chicks reared in constant darkness for 6 hr, compared with birds reared in any other light conditions examined. It should be noted that the light intensities of the red and blue LED lights used in the present study were relatively low, compared with that of the medium and high intensities of the white LED lights, due to limitations of the maximum light intensity of the red LED lights. It is therefore possible that choroidal IL-6 gene expression might be differentially affected under higher intensity red and blue LED lighting conditions. *Wang et al., 2018* previously reported that continuous exposure to blue LED light (435 lux) for 5 days caused a significant hyperopic shift in refraction in both control and form deprived eyes, but had no significant effect on axial length in either control or form deprived eyes (although a trend toward a decrease in axial length was observed in both control and form deprived eyes). Since refraction was significantly affected in blue LED light-reared chicks, but axial length was only minimally affected, we suspect that continual exposure to blue light may have affected other ocular parameters (such as corneal curvature) that would have a significant impact on refraction. We predict that choroidal IL-6 expression is involved in the choroidal and scleral remodeling processes at the posterior pole of the eye that result in changes in vitreous chamber elongation, as opposed to having effects on the anterior segment of the eye. Since IL-6 gene expression was not affected by short-term exposure to red or blue LED light, we predict that exposure to red or blue LED light would have no effect on scleral remodeling, vitreous chamber depth, or axial length, under the conditions used in the present study. Further studies, using higher intensity LED lights,

together with additional ocular growth measurements (such as corneal curvature, anterior chamber depth, and lens thickness) are necessary to fully elucidate the role of red and blue LED light on choroidal IL-6 expression, ocular growth, and refraction.

Protein levels of IL-6 were also significantly increased in choroids of recovering eyes following 6 hr of recovery, but were not significantly different following 24 hr of recovery. We suspect that IL-6 protein is released from choroids and either enters the circulation or adjacent ocular tissues shortly after its synthesis as has been described for IL-6 in skeletal muscle (*Steensberg et al., 2000*).

Clinical studies have previously suggested an association between inflammation and the progression of myopia. A higher incidence of myopia has been reported among patients with inflammatory diseases such as type 1 diabetes, uveitis, juvenile chronic arthritis, or systemic lupus erythematosus compared to those without inflammatory diseases (*Lin et al., 2016*; *Fledelius et al., 2001*; *Herbort et al., 2011*; *Kamath et al., 2013*). *Herbort et al., 2011* hypothesized that anatomic changes of the eye due to myopia lead to fragility of the choriocapillaris and RPE, predisposing myopic eyes to inflammatory ocular conditions, thereby suggesting that ocular inflammation is a consequence, rather than a cause of myopia. However, our observed elevation in choroidal IL-6 gene and protein expression in response to recovery/myopic defocus does not appear to be due to a general inflammatory response, as other pro-inflammatory cytokines, IFN-γ, and TNF-α, were unaffected by visual manipulations. IL-1B was significantly elevated following 6 hr of recovery, but was unaffected at 1.5 and 3 hr of recovery, suggesting that changes in IL-1B gene expression are downstream to that of IL-6, as has been demonstrated for Il-6 and other plasma cytokines following strenuous exercise (*Ostrowski et al., 1999*), cancer (*Tilg et al., 1994*), and infection (*Jordan et al., 1995*).

The finding that myopic defocus significantly upregulated choroidal IL-6 transcription, but hyperopic defocus had only a modest effect at downregulating choroidal IL-6, may be related to the observation that myopic defocus is a more potent stimulus than hyperopic defocus at causing changes in choroid thickness, with regard to the duration of lens wear needed to elicit changes in choroidal thickness, as well as the duration of the response to lens wear (*Zhu et al., 2005*). If IL-6 is involved in mediating changes in choroidal thickness, we would predict that myopic defocus-induced increases in choroidal IL-6 gene and protein expression would induce changes in choroidal thickness that would outweigh and outlast the minor reduction in IL-6 expression induced by brief periods of hyperopic defocus.

If choroidal IL-6 gene expression is causally related to ocular changes associated with recovery from myopia, then agents known to block recovery should also block choroidal IL-6 upregulation. To test this, we evaluated choroidal IL-6 gene expression in recovering eyes following intravitreal application of the NOS inhibitor, L-NAME. L-NAME has previously been shown to prevent choroidal thickening and disinhibit scleral proteoglycan synthesis during recovery from induced myopia and in response to positive lens wear (*Nickla et al., 2006*). We found that intravitreal treatment with L-NAME significantly attenuated the recovery-induced increase in choroidal IL-6 gene expression, indicating that nitric oxide either directly or indirectly regulates choroidal IL-6 gene expression. Treatment of isolated choroids with the NO donor, PAPA-NONOate stimulated IL-6 gene, and protein expression, confirming that choroidal IL-6 is upregulated by NO. NO has been reported to stimulate IL-6 production in skeletal myocytes (*Makris et al., 2010*), human blood mononuclear cells (*Siednienko et al., 2011*), and kidney epithelial cells (*Demirel et al., 2012*) by activating ERK1/2 and p38 MAPK-signaling pathways. We also found that p38 MAPK signaling was crucial for the PAPA-NONOate-mediated activation of IL-6 in chick choroids as incubation with the p38 MAPK inhibitor, SB203580, abolished PAPA-NONOate-stimulation of IL-6 gene expression.

The concentration of retinal dopamine has been implicated in the regulation of eye growth. Daytime levels of retinal dopamine were shown to be reduced in form deprived chick (*Stone et al., 1989*) and monkey eyes (*Iuvone et al., 1989*). Furthermore, intravitreal injections of apomorphine (*Stone et al., 1989*; *Iuvone et al., 1991*) or dopamine (*Gao et al., 2006*) prevented the development of deprivation myopia. Interestingly, *Nickla et al., 2013* demonstrated that the dopamine agonist, quinpirole, prevented the development of myopia via a nitric oxide-dependent mechanism, indicating that dopamine acts upstream of NO in the signal cascade mediating ocular growth inhibition. In view of our results demonstrating a role for nitric oxide in the regulation of choroidal IL-6 expression, we suspect that dopamine and nitric oxide are located upstream of choroidal IL-6 in the retina-to-scleral signaling cascade.

**Figure 10.** Proposed role of IL-6 in the retina-to-sclera signaling cascade. Myopic defocus initiates a series of signaling events in the retina potentially involving dopamine, nitric oxide synthase (NOS), and nitric oxide (NO) (as well as other mediators not shown). Nitric oxide, synthesized in the retina and/or in the choroid by NOS from L-arginine (L-arg), stimulates choroidal expression of IL-6 via a p38 MAPK-dependent mechanism. Choroidal IL-6, in turn, potentially coordinates many of the features of the choroidal response to myopic defocus including: (1) increased synthesis of hyaluronan synthase 2 (HAS2) and hyaluronic acid (HA), (2) increased synthesis of vascular endothelial growth factor (VEGF), and (3) increased cell proliferation, which result in choroidal thickening, increased vascular permeability, and increased retinaldehyde dehydrogenase 2 (RALDH2), respectively. These choroidal changes lead to the production of scleral growth regulators, such as all-*trans*-retinoic acid (atRA), that regulate scleral remodeling, such as decreased scleral proteoglycan (PG) synthesis (in chicks) to result in a slowing of ocular growth and recovery from myopia.

In the present study, since L-NAME was delivered intravitreally and NO was delivered directly to the choroid (via PAPA-NONOate), it is unclear as to the cellular source of NO generated in response to myopic defocus that is responsible for IL-6 upregulation. NO and nNOS (NOS1) have been detected in virtually all types of retinal neurons, in the RPE, and in several cell types in the chick choroid (*Fischer and Stell, 1999*; *Eldred, 2000*). Our finding suggests that treatment of isolated choroids with the NOS substrate, L-arginine, in the presence of 50 mM KCl can similarly upregulate choroidal IL-6 transcription that suggests the endogenous sources of nitric oxide within the choroid have the potential to regulate IL-6 mRNA and protein synthesis. It is also unclear as to the cellular source of choroidal IL-6 in response to myopic defocus. Our immunohistochemical evaluation of IL-6 protein distribution indicated that IL-6 was present as discrete puncta in the RPE, choroidal vascular endothelial cells, and extravascular stromal cells. Any of these cells, as well as myeloid and lymphoid cells, could be the source of visually induced IL-6. However, IL-6-positive choroidal cells identified in the present study may indicate internalization of IL-6 following synthesis and secretion by neighboring cells via a paracrine signaling mechanism, and not necessarily the site of IL-6 synthesis.

Atropine has been shown to prevent experimentally induced myopia in chicks and inhibits myopia development in some children when applied topically (*Chia et al., 2012*). We therefore evaluated choroidal IL-6 gene expression following in vivo and in vitro treatment with atropine. We found that a single intravitreal injection of atropine to chicks undergoing form deprivation-induced myopia stimulated choroidal IL-6 gene expression. Moreover, incubation of isolated choroids with 0.1% atropine also stimulated choroidal IL-6 gene and protein expression, suggesting that atropine acts directly on the choroid to stimulate IL-6 synthesis.

The results of the present study indicate that choroidal IL-6 is a likely player in the retina-to-sclera signaling cascade underpinning visually guided ocular growth (*Figure 10*). Of great interest, therefore, are the upstream regulators of choroidal IL-6 gene transcription that relay visual signals, initiated in the retina, to the choroid to bring about changes in IL-6 protein synthesis. IL-6 transcription has been shown to be regulated by a number of factors, including steroid hormones, glucocorticoids, TNF-α, interferon-γ, Il-1b, epidermal growth factor, and amphiregulin (*Luo and Zheng, 2016*; *Bersinger et al., 2011*; *Lisi et al., 2010 Mahtouk et al., 2005*), and some of these mediators have been implicated in myopia development (*Dong et al., 2019*; *Dong et al., 2020*; *Ding et al., 2018*; *Gong et al., 2015*). Investigations into the retinal and/or RPE expression of these and other factors, known to regulate IL-6 mRNA and protein synthesis, may provide new insights into the visually driven signaling cascade responsible for emmetropization and myopia development.

## Conclusions

In the present study, we report that myopic defocus, either in eyes recovering from induced myopia, or in eyes treated with +15 D spectacle lenses, stimulates IL-6 mRNA and protein synthesis in the chick choroid. The ramifications of increased choroidal IL-6 synthesis are unclear. In the context of ocular

growth control, it appears that choroidal IL-6 is associated with a slowing of eye growth, as it is upregulated in recovering eyes (when eyes are decelerating their rate of elongation) and in myopic eyes treated with atropine, an agent known to inhibit vitreous chamber elongation and myopia. Moreover, IL-6 mRNA is downregulated in recovering eyes treated with L-NAME, a compound known to inhibit recovery and increase scleral proteoglycan synthesis and ocular elongation. Treatment of isolated sclera with IL-6 had no effect on scleral proteoglycan synthesis (*Figure 3—figure supplement 1*), indicating that additional downstream mediators, most likely derived from the choroid, are responsible for regulating the scleral changes associated with recovery.

On the other hand, studies have shown that IL-6 has a major role in the pathology of uveitis, glaucoma, retinal vein occlusion, macula edema, and diabetic retinopathy (*Zahir-Jouzdani et al., 2017*). IL-6 induces ocular inflammatory responses often leading to the breakdown of the blood ocular barrier, angiogenesis, increased vascular permeability, and choroidal neovascularization. Based on the results of the present study, it is possible that myopic defocus in humans, as a result of uncorrected myopia, may cause elevated choroidal IL-6 which could predispose individuals to one or more of the above serious ocular complications.

The identification of small molecule or biological approaches to manipulate choroidal IL-6 concentrations will elucidate the role of choroidal IL-6 in postnatal ocular growth, as well as in a variety of ocular conditions.

## Materials and methods

**Key resources table**

| Reagent type (species) or resource | Designation | Source or reference | Identifiers | Additional information |
|---|---|---|---|---|
| Antibody | Anti-chick IL-6 (Rabbit polyclonal) | Bio-Rad Laboratories | Cat#: AHP942Z RRID: AB_2127753 | IF (1:20) |
| Sequence-based reagent | Chicken IL-6 Taqman Gene Expression Assay | Thermo Fisher Scientific | Gg03337980_m1 unlabeled PCR primers and FAM-labeled TaqMan probe | |
| Sequence-based reagent | Chicken interferon γ TaqMan Gene Expression Assay | ThermoFisher Scientific | Gg03348618_m1 unlabeled PCR primers and FAM-labeled TaqMan probe | |
| Sequence-based reagent | Chicken IL-1β TaqMan Gene Expression Assay | Thermo Fisher Scientific | Gg03347154_g1 unlabeled PCR primers and FAM-labeled TaqMan probe | |
| Sequence-based reagent | Chicken TNF-$\alpha$ (LITAF) TaqMan Gene Expression Assay | Thermo Fisher Scientific | Gg03364359_m1 unlabeled PCR primers and FAM-labeled TaqMan probe | |
| Sequence-based reagent | Chicken GAPDH TaqMan Gene Expression Assay | Thermo Fisher Scientific | Gg03346982_m1 unlabeled PCR primers and FAM-labeled TaqMan probe | |
| Peptide, recombinant protein | Chicken IL-6 | Bio-Rad Laboratories | Cat#: PAP003 | |
| Commercial assay or kit | High capacity RNA-to-cDNA kit | Applied Biosystems | Cat#: 4388950 | |
| Commercial assay or kit | DNase treatment & removal kit | Invitrogen | Cat#: AM1906 | |
| Chemical compound, drug | SB 203580 | Sigma-Aldrich | Cat#: S8307 | |
| Chemical compound, drug | Atropine (Sulfate Salt) | Sigma-Aldrich | Cat#: A-0257 | |
| Chemical Compound, drug | L-NAME | Sigma-Aldrich | Cat#: N5751 | |
| Chemical compound, drug | PAPA-NONOate | Cayman Chemical | Cat#: 82,140 | |
| Other | DAPI stain | Invitrogen | Cat#: D3571 | (5 µg/ml) |

## Animals

Animals were managed in accordance with the ARVO Statement for the Use of Animals in Ophthalmic and Vision Research, with the Animal Welfare Act, and with the National Institutes of Health Guidelines. All procedures were approved by the Institutional Animal Care and Use Committee of the University of Oklahoma Health Sciences Center (protocol # 20-092H ). White Leghorn male chicks (*Gallus gallus*) were obtained as 2-day-old hatchlings from Ideal Breeding Poultry Farms (Cameron, TX). Chicks were housed in temperature-controlled brooders with a 12 hr light/dark cycle and were given food and water ad libitum. At the end of experiments, chicks were euthanized by an overdose of isoflurane inhalant anesthetic (IsoThesia; Vetus Animal Health, Rockville Center, NY), followed by decapitation.

## Visual manipulations

FDM was induced in 3–4-day-old chicks by applying translucent plastic goggles to one eye, as previously described (*Rada et al., 1991*). The contralateral eyes (left eyes) of all chicks remained untreated and served as controls. Chicks were checked daily for the condition of the goggles. Goggles remained in place for 10 days, after which time the goggles were removed and chicks were allowed to experience unrestricted vision (recover) for up to 4 days. When multiple time points were assessed in one experiment, chicks were randomly assigned to groups for each time point.

Lens-induced myopia and hyperopia were induced via the application of +15 and –15 D lenses to 3–4-day-old chicks. Lenses were fashioned from PMMA hard contact lenses (12 mm diameter, 8 mm base curve, Conforma Labs, Inc, Norfolk, VA) that were mounted onto nylon washers for support using optical adhesive (Norland Products, Inc, Cranbury, NJ). A velcro ring was glued to the back of the nylon washer for mounting around the chick's right eye using cyanoacrylate adhesive. Lenses remained in place for up to 24 hr.

For light intensity experiments, cages (24″×24″×16″, L×W×H, respectively) were fitted with Multi-color (RGB) and White LED strip lights at the top surface of the cage and light intensity was controlled using a wireless RF remote (Super Bright LEDs, Inc, St. Louis, MO). Light intensity (5–3150 lux) was measured using a light meter (Datalogger Model 401036, Extech Instruments, Nashua, NH) at a distance of 8 cm from the bottom of the cage (approximate eye-level of chicks). Spectral peaks of LED light sources were obtained using a luminous flux tester spectrometer (OHSP-350, Hangzhou Hopoo Light & Color Technology Co, Zhejiang, China) (*Figure 4—figure supplement 1*). Chicks were randomly assigned to white, red, or blue light and housed in LED cages for 6 hr (9:30 a.m. to 3:30 p.m.). A separate group of chicks was kept in complete darkness for 6 hr.

## Intravitreal injections

Injections were delivered using a NanoFil-100 syringe with a 26G needle (World Precision Instruments, Sarasota, FL) under isoflurane ( 0.8% in $O_2$; IsoThesia; Vetus Animal Health, Rockville Center, NY) inhalation anesthesia at a flow rate of 0.4 L/min using an Isoflurane Anesthesia machine for veterinary use only (Ohmeda Anesthesia Service and Equipment, Inc, Atlanta, GA). Following removal of the occluder, the sclera was exposed by retracting the eyelids with a handmade ocular speculum and injections were delivered through the sclera at the superior margin of the globe, just outside of the scleral ossicles, after cleaning the eyelids and surround area with 70% alcohol. Injections consisted of L-NAME (Sigma Chemical Co, St. Louis, MO) (a 30 µl injection containing 16.2 µmol of L-NAME in 0.9% saline), 30 µl of 0.9% saline (vehicle for L-NAME) (*Nickla and Wildsoet, 2004*), atropine sulfate (Sigma Chemical Co) (a 20 µl injection containing 240 nmol of atropine sulfate in phosphate-buffered saline, PBS), and 20 µl of PBS (vehicle for atropine sulfate) (*Carr and Stell, 2016*). The needle remained in place for 15 s before slowly withdrawing it from the eye and an ophthalmic antibiotic ointment (Vetropolycin, Pharmaderm, Melvill, NY) was applied to the eye. In some cases, the occluders were replaced prior to awakening from the anesthesia.

## Photorefraction

Measurements of the refractive state were performed on anesthetized chicks following 10 days of form deprivation or while wearing +15 D and –15 D lenses without cycloplegia using infrared photo-refraction (*Schaeffel et al., 2004*), performed at a sampling frequency of 62 Hz from a distance of 1 m in a dim room (ambient illuminance about 0.5 lux) (retinoscope and software obtained from Steinbeiss Transfer Centre for Biomedical Optics, Tuebingen, Germany).

## Tissue preparation

Chicks were euthanized by an overdose of isoflurane inhalant anesthetic (IsoThesia; Vetus Animal Health) following 10 days of form deprivation (day 0 recovery), after various time points of recovery, following lens wear, or light exposure. Eyes were enucleated and cut along the equator to separate the anterior segment and posterior eye cup. Anterior tissues were discarded, and the vitreous body was removed from the posterior eye cups. An 8 mm punch was taken from the posterior pole of the chick eye using a dermal biopsy punch (Miltex Inc, York, PA). Punches were located nasal to the exit of the optic nerve, with care to exclude the optic nerve and pecten oculi. With the aid of a dissecting microscope, the retina and majority of RPE were removed from the underlying choroid and sclera with a drop of PBS (3 mM dibasic sodium phosphate, 1.5 mM monobasic sodium phosphate, 150 mM NaCl, and pH 7.2) and gentle brushing. For microarray, TaqMan real-time PCR, and ELISA assays, choroids were separated from the sclera using a small spatula, placed in 2 ml screw cap tubes, and snap frozen in liquid nitrogen and stored at – 80°C. For immunolabeling experiments, choroids with sclera still attached were placed into a 48-well flat-bottom plate (Corning Inc, Corning, NY). A small amount of RPE was left on the choroids to discriminate between the RPE and scleral side of the tissue. The tissues were then fixed with 4% paraformaldehyde (stock solution freshly prepared) in PBS O/N at 4°C.

## Immunolabeling of chick choroids

Punches (5 mm) containing retina, RPE choroid, and sclera were obtained from the posterior poles of control and recovering chick eyes, fixed in neutral-buffered formalin, and embedded in paraffin, and sections were obtained. Tissue sections of posterior ocular tissues were deparaffinized through a graded series of xylenes and ethanol and rinsed in PBS. Slides were then transferred to a Coplin jar containing citrate buffer, freshly prepared from a 10× concentrate (Thermo Fisher Scientific), and incubated in a rice steamer (Black & Decker, Towson, MD) for 40 min for antigen retrieval. Slides were then cooled for 30 min, washed 2× in PBS and then incubated for 30 min at room temperature (RT) in incubation buffer that consisted of 2% BSA (Sigma Chemical Co) and 0.2% Triton X-100 in PBS. Sections were incubated overnight at 4°C with rabbit anti-chick IL-6 (Bio-Rad Laboratories, Inc, Hercules, CA) diluted 1:20 in incubation buffer. For negative controls, tissue sections were incubated in 25 µg/ml nonimmune rabbit immunoglobulin (Sigma Chemical Co) instead of the IL-6 antibody. Additional pre-absorption controls were performed in which the anti-IL-6 antibody was incubated overnight at 4°C with a tenfold molar excess of recombinant chicken IL-6 (1.67 µM; Bio-Rad Laboratories, Inc) before immunolabeling fixed sections of chick ocular tissues. Following overnight incubation with the primary antibody, sections were rinsed in PBS, and incubated for 30 min at RT in 5 µg/ml of goat anti-rabbit Alexa Fluor 488 (Thermo Fisher Scientific, Richardson, TX). Sections were rinsed in PBS and then incubated for 10 s at RT with 0.0005% DAPI nuclear stain, followed by a final rinse in PBS. Coverslips were mounted onto the slides with Prolong Gold Antifade reagent containing DAPI (Thermo Fisher Scientific), and the immunolabeled sections were examined under an Olympus Fluoview 1000 laser-scanning confocal microscope (Center Valley, PA).

## Microarray

Choroids were isolated from 10 normal chick eyes (n=5 chicks) and from control and treated chicks eyes following 6 hr of recovery from 10 days of prior form deprivation-induced myopia (n=5 chicks) and kept at –80°C until processed. Choroids were shipped on dry ice to the Microarray Core Facility at the University of Tulsa (Tulsa, OK). When processing began the samples were moved to a container of liquid nitrogen. The choroids were pulverized using a frozen 1.5 ml disposable pestle. Immediately following pulverization the samples were immersed in 300 µl of Ambion TriReagent (Applied Biosystems, Foster City, CA) solution and homogenized for 90 s with a Pellet Mixer. An additional 700 µl of TriReagent was pipetted into the sample after homogenization. Incubation of samples occurred for 5 min using a 1.5 ml microfuge tube shaker at RT. The samples were then transferred to pre-spun Phase Lock Gel Heavy 2 ml Gel tubes (5 Prime Inc, Gaithersburg, MD). 200 µl of chloroform was added to each sample, inverted 12 times, and incubated at RT for 5 min. The samples were then spun at 2°C for 20 min. 500 µl supernatant was poured into 2 ml round-bottom tubes. These tubes were placed into the Qiagen Qiacube robotic workstation and cleaned using the RNeasy Lipid Tissue Mini

Kit (Qiagen, Redwood City, CA). The samples were eluted in 50 µl of molecular biology water. The samples were also split into two 25 µl aliquots to ensure sample safety.

Following RNA isolation, the samples were quantified using a NanoDrop 1000 spectrophotometer (Thermo Fisher Scientific). The initial average sample concentration ranged from 20 to 88 ng/µl. The initial RNA 260/280 ratios were between 1.8 and 2.0 with the 260/230 ratios between 0.8 and 1.9. Precipitation of one aliquot of RNA was performed to increase the sample concentration and purity. This procedure was performed by addition of 2.5 volumes of ice-cold 100% EtOH, 1/10 of 3 M ammonium acetate, and 1 µl of glycogen at 5 ng/µl. The samples were incubated at –20°C overnight. The samples were spun at 4°C for 30 min to pellet the RNA. The supernatant was removed and the pellet was washed with ice-cold 80% EtOH to remove the remaining salt. The EtOH was aspirated off and the pellet was dried at RT for 5 min. Molecular biology water was used to re-suspend the RNA pellet. The amount of water used was calculated to bring the sample concentration to between 58 and 133 ng/µl. After precipitation, the 260/280 ratios are between 2.0 and 2.1 and the 260/230 ratios are between 1.8 and 2.1. 150 ng of each sample was processed with the Affymetrix 3′ IVT Express Kit (Thermo Fisher Scientific).

## Microarray data analysis

Gene expression was analyzed on an Affymetrix Gene Chip Chicken Genome Array containing 38,535 probes. Slides were scanned by an Agilent Microarray Scanner (Agilent Technologies) and data were extracted by Feature Extraction software 10.7 (Agilent Technologies). The raw data were normalized by the Robust Multichip Average method of normalization (*Bolstad et al., 2003*; *Irizarry et al., 2003a*; *Irizarry et al., 2003b*). The data were grouped into those of normal choroids, recovering choroids, and contralateral control choroids and analyzed using the Method of Moments one-way ANOVA (*Eisenhart, 1947*) and the Benjamini-Hochberg Step-up procedure for the false discovery rate (FDR) (*Benjamini and Hochberg, 1995*). Upregulated or downregulated genes were identified by at least twofold changes and the genes with a p-value (adjusted for FDR) below 0.05 were considered statistically different and identified as differentially expressed genes.

## TaqMan quantitative PCR (RT-quantitative PCR)

Choroids were isolated from individual pairs of control and treated eyes and snap frozen in liquid nitrogen. Total RNA was isolated using TRIzol reagent (Thermo Fisher Scientific) followed by DNase treatment (DNA-free, Applied Biosystems) as described previously (*Summers et al., 2016*). RNA concentration and purity were determined via the optical density ratio of 260/280 using a Nanodrop ND-1000 spectrophotometer and stored at − 80°C until use. cDNA was generated from DNase-treated RNA using a High Capacity RNA to cDNA Kit. Real-time PCR was carried out using a Bio-Rad CFX 96. 20 µl reactions were set up containing 10 µl of TaqMan 2× Universal Master Mix (Applied Biosystems), 1 µl 20 × 6-carboxyfluorescein (FAM)-labeled Assay Mix (Applied Biosystems), and 9 µl of cDNA. Each sample was set up in duplicate with specific primers and probed for chicken IL-6 (assay ID number Gg03337980_m1), chicken interferonγ (INFG, assay ID number Gg03348618_m1), chicken IL-1β (IL1β, assay ID number Gg03347154_g1), chicken TNF-α (LITAF, assay ID number Gg03364359_m1), and the reference gene chicken GAPDH (assay ID number Gg03346982_m1) (Thermo Fisher Scientific). The PCR cycle parameters were an initial denaturing step at 95°C for 10 min followed by 45 cycles of 95°C for 15 s and 60°C for 1 min. Normalized gene expression was determined by the ΔΔc(t) method (*Livak and Schmittgen, 2001*) using Bio-Rad CFX Manager version 3.1 and reported values represent the average of duplicate samples.

## IL-6 protein measurements

Punches (8 mm) of chick choroids were rinsed in ice-cold PBS (0.01 M, pH 7.2) and homogenized in 300 µl of PBS on ice (Omni Tip, Omni International, Kennesaw, GA). The resulting suspension was sonicated with an ultrasonic homogenizer (Pulse 150, Benchmark Scientific, Edison, NJ) and subjected to two freeze-thaw cycles to further break the cell membranes. Homogenates were then centrifugated for 5 min at 5000×*g*. Following centrifugation, pellets were discarded and the supernatants stored at ≤− 20°C. IL-6 was measured on duplicate samples using a commercially available chicken IL-6 ELISA Kit (Aviva Systems Biology, Corp, San Diego, CA) according to the manufacturer's instructions.

Protein concentrations in choroidal lysates were determined on duplicate samples by Bradford assay. Reported values represent the average of duplicate samples.

## Organ culture

Choroids were isolated from eyes from adult chicken heads (Animal Technologies, Inc, Tyler, TX) as described above and placed in 48-well plates containing 300 µl culture medium (1:1 mixture of Dulbecco's modified Eagle's medium [DMEM] and Ham's F12 containing streptomycin [0.1 mg/ml], penicillin [100 units/ml], and gentamicin [50 µg/ml]) in the presence of the NO donor, PAPA-NONOate (0.5–5 mM in culture medium; Cayman Chemical, Ann Arbor, MI), the p38 MAPK inhibitor, SB203580 (10 µM; Sigma-Aldrich), atropine sulfate ( 0.1%; Sigma-Aldrich), or culture medium alone in a humidified incubator with 5% $CO_2$, overnight at 37°C. Following incubation, choroids were snap frozen and RNA isolated for TaqMan real-time PCR assays, and medium harvested and frozen for IL-6 ELISA assays.

## Scleral sulfated glycosaminoglycan synthesis

The posterior hemispheres of eyes of FD chicks (=0 days of recovery), or from eyes from chicks recovering from FD myopia for 1–20 days and contralateral controls were obtained and one 5 mm tissue punch was excised from the posterior sclera of control and treated eyes using a dermal punch (Miltex Instrument Co). All retina, RPE, choroid, vitreous, pectin, and muscle were gently cleaned from each sclera punch. Scleral punches were initially placed into wells of a 96-well culture plate with 50 µl of N2 medium Ham's F-12/DMEM containing 1× N2 supplement (Stem Cell Technologies, Vancouver, BC) until all sclera were obtained. Scleral punches were then transferred to N2 medium containing $^{35}SO_4$ (100 µCi/ml; New England Nuclear, MA) and incubated for 3 hr at 37°C. Radiolabeled scleral punches were digested with proteinase K (protease type XXVIII, Sigma Chemical Co) (0.05% w/v in 10 mM EDTA, 0.1 M sodium phosphate, and pH 6.5) overnight at 60°C. $^{35}SO_4$-labeled glycosaminoglycans (GAGs) were precipitated by the addition of 0.5% cetylpyridinum chloride (CPC) in 0.002 M $Na_2SO_4$ in the presence of unlabeled carrier chondroitin sulfate (1 mg/ml in $dH_2O$). The samples were incubated for 30 min at 37°C and precipitated GAGs were collected on Whatman filters (GF/F) using a Millipore 12-port sampling manifold as previously described (*Rada et al., 1992*). Radioactivity was measured directly on the filters by liquid scintillation counting.

## Statistics

Sample sizes were calculated using G*Power 3.1.9.2 using two-tailed tests with an α=0.05, and an effect size determined by group means and standard deviations previously published by this lab and others (*Rada et al., 1991*; *Wallman and Adams, 1987*). All experiments were repeated at least one time, and sample sizes and results reported reflect the cumulative data for all trials of each experiment. All data were subjected to the D'Agostino and Pearson test to test the normality of the data. Data that passed the D'Agostino and Pearson test were subjected to parametric analyses. Parametric analyses between two groups were made using paired or unpaired Student's t-tests, and multiple comparisons were analyzed using a one-way ANOVA followed by a Bonferroni correction. Data that failed the D'Agostino and Pearson test, or had a sample size too small for the D'Agostino and Pearson normality test were subjected to nonparametric analyses. Nonparametric tests between two groups were made using the Wilcoxon signed-rank test for matched pairs, the Mann-Whitney U-test, or the Kruskal-Wallis test for multiple comparisons (GraphPad Prism 5, La Jolla, CA). Results were considered significant with p-value≤0.05.

## Acknowledgements

This work was supported by NIH grant R01EY09391 (JAS) and by NIGMS COBRE Grant P30GM122744 (Ma, J-X., PI). The authors would like to thank Dr. Frederick (Kris) Miller (Department of Cell Biology, University of Oklahoma Health Science Center) and Dr. Randle Gallucci (Department of Pharmaceutical Sciences, University of Oklahoma Health Science Center) for their helpful discussions and suggestions.

## Additional information

### Funding

| Funder | Grant reference number | Author |
|---|---|---|
| National Eye Institute | EY09391 | Jody Ann Summers |
| National Institute of General Medical Sciences | P30GM122744 | Jody Ann Summers |
| National Institutes of Health | R01EY09391 | Jody A Summers |

The funders had no role in study design, data collection and interpretation, or the decision to submit the work for publication.

### Author contributions

Jody A Summers, Conceptualization, Data curation, Formal analysis, Funding acquisition, Investigation, Methodology, Project administration, Supervision, Visualization, Writing – original draft; Elizabeth Martinez, Data curation, Formal analysis, Investigation, Methodology

### Author ORCIDs

Jody A Summers (iD) http://orcid.org/0000-0001-8847-7812

### Ethics

Animals were managed in accordance with the ARVO Statement for the Use of Animals in Ophthalmic and Vision Research, with the Animal Welfare Act, and with the National Institutes of Health Guidelines. All procedures were approved by the Institutional Animal Care and Use Committee of the University of Oklahoma Health Sciences Center. (protocol # 20-092-H).

### Decision letter and Author response

Decision letter https://doi.org/10.7554/eLife.70608.sa1
Author response https://doi.org/10.7554/eLife.70608.sa2

## Additional files

### Supplementary files

• Transparent reporting form

• Supplementary file 1. Affymetrix microarray analysis of gene expression in control and recovering chick choroids.

### Data availability

All data generated or analyzed during this study are included in the manuscript and supporting files.

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
