## [Decision Letter]

**Acceptance summary:**

Myopia is an ocular disorder of increasing concern to human individuals and health-care systems. Usually it is due to excessive elongation of the optic axis of the eye during the ages of most rapid growth, causing images of distant objects to be blurred at the retinal photoreceptors. Despite extensive epidemiological and animal studies in the past several decades, the underlying causal mechanisms remain poorly known, and therapeutic options are limited. Therefore, further discovery of new candidate mechanisms, drug targets and drugs for inhibiting the onset and progression of myopia is urgently needed. This paper links the upregulation of IL-6 expression to axial elongation and myopic defocus experienced by the retina. This manuscript is an important contribution to the elucidation of the mechanistic underpinnings of myopia.

**Decision letter after peer review:**

Thank you for submitting your article "Visually Induced Changes in Cytokine Production in the Chick Choroid" for consideration by *eLife*. Your article has been reviewed by 3 peer reviewers, and the evaluation has been overseen by a Reviewing Editor and Anna Akhmanova as the Senior Editor. The following individual involved in review of your submission has agreed to reveal their identity: William K. Stell (Reviewer #1).

Essential Revisions (for the authors):

The reviewers were excited about the novel finding connecting in juvenile chicken models, the pro-inflammatory cytokine, interleukin-6 (IL-6) – synthesized and released in the choroid – to the regulation of axial elongation and refraction of the eye.

1) Most of the issues revolved around the connection between NO signaling and IL-6. The reviewers would like either new experiments to better substatiate this connection or perhaps consider removing this concept altogether. The reviewers did not come to a consensus on this point but overall, they all agreed that with changes, publication in *eLife* is warranted. Therefore, please consider both options.

2) There are significant changes to the writing required. Please see each point of all three reviews. In addition, higher magnification microscopy is required.

*Reviewer #1 (Recommendations for the authors):*

It would be extraordinarily helpful for you to create a schematic representation of your concluding hypothesis – that is, how you think IL-6 fits into the signalling system or network that transforms visual information from the retina into regulation of scleral 'growth' and axial elongation.

*Reviewer #2 (Recommendations for the authors):*

This is a solid experimental study of the role of IL-6 in the control of the response of chicken eyes to myopic defocus in two paradigms – that in which, after the removal of the diffusers, in eyes made myopic through form-deprivation, the rate of axial elongation slows, and in the paradigm in which myopic defocus imposed with positive lenses similarly slows axial elongation.

There is, however, a general issue in the terminology used in the presentation of the results that causes confusion. For example, in Figure 1, it is not clear to me what treated eyes are. The authors need to carefully distinguish between untreated but age-matched control eyes, contralateral control eyes, and eyes that have developed myopia and the are baseline for looking at the recovery process.

Figure 1 shows IL-6 immunolocalisation in treated and recovering eyes. I am afraid that the magnifications are too low to clearly see localisation to the structures mentioned, or to see any changes. An image of general cell staining would help in for orientation.

Figure 2 shows that IL-6 mRNA expression increases during recovery by over 10-fold. I cannot see any evidence of a corresponding change in the immunohistochemistry in Figure 1. Why?

Figure 3 shows a transient (around 4 day) increase in IL-6 mRNA expression during the recovery process, and major changes in expression of TNF-α.

Figure 4 confirms that IL-6 gene expression is down-regulated in the dark, and shows that a range of lighting conditions restores expression.

Figure 5 shows that IL-6 gene expression is increased by imposed myopic defocus, but that imposed hyperopic defocus has much smaller, possibly insignificant effects. Partially corresponding changes were seen in scleral proteoglycan synthesis.

Figure 6 shows that L-NAME reduces the changes in expression during recovery, implicating NO in the pathway. Figure 7 uses an NO-donor on isolated choroids to confirm this effect.It also shows that MAPK pathways may be involved.

Figure 8 shows that atropine, which is known to slow axial elongation, also increases IL-6 expression in the choroid.

One of the intriguing findings of the study is that atropine stimulates IL-6 expression in isolated choroids, suggesting a direct effect. Since the site and mode of action remains controversial despite its now wide-spread use in controlling myopia progression, this is an important lead for future experimentation.

There are also some excessive statements made in the Introduction that need correction.

Line 39: The estimates of the future prevalence of myopia made by Holden et al., are only predictions. They might become "expectations" if there was general acceptance of the underlying methodology, but I believe that the model on which the predictions are based is fundamentally flawed. Unless the authors have carefully considered the methodology, and find it convincing, the authors should reword this section. This problem can be fixed by changing the word "expected" to "predicted."

Line 41-42: The statement that "the cause of human myopia is not understood" cannot be justified. This statement does not reflect the current state of the literature, although it may be that the authors are not familiar with the literature on human myopia. The simplest way to solve this problem is simply to drop the statement. Otherwise, the authors will need to become more familiar with the literature, which suggests that at least two major causes have been identified, namely intensive education, perhaps captured by nearwork, and deprivation of time outdoors. Both are particularly common in East and Southeast Asia where it is well-known that there is an epidemic of myopia.

While I do not want to dispute the next statement, that work on animal models has provided valuable insights into the role that the visual environment plays in control of eye growth, it is not clear how the defocus controls studied in this paper are involved in the effects of nearwork, if they in fact are. The situation is clearer with deprivation of time outdoors, because it links to a well-described role of decreased dopamine release in the development of myopia in the animal models. This is important for the current paper because Nickla and colleagues have shown that dopaminergic agents affect choroidal thickness, suggesting a rather obvious set of further experiments.

Line 45: Normal visual input is not restored by removal of the occluder. Rather, a myopically defocused image is imposed. This would still lack much of the high spatial frequency content eliminated by the occlude, but it is clearly sufficient to restore some normality to control of eye growth.

The authors raise the potentially interesting possibility that since myopic defocus increases IL-6 expression, that there is over-expression in myopia eyes, presumably even in adult eyes that no longer respond actively to growth signals. They imply that this could contribute to at least some of the long-term consequences of high myopia in particular. The qualification that needs to be added is that it does not seem that correction reduces the pathological consequences of myopia, but given the long-term nature of the processes in humans, and the fact that correction is rarely constant, it remains an interesting suggestion.

*Reviewer #3 (Recommendations for the authors):*

Several deficiencies must be addressed before the paper is published:

1. Lines 41-42. I would disagree that "the cause of myopia in humans is not understood." Although many environmental factors may influence refractive eye development, the overwhelming evidence suggest that the cause of myopia is excessive exposure to nearwork and hyperopic optical defocus.

2. Lines 43-44. Grammar. "role of the visual environment [in] ocular growth control."

3. Lines 44-46. "Deprivation of form vision, through the use of visual "occluders" or "goggles" results in accelerated ocular growth and the development of myopia within a matter of days in chicks, tree shrews, guinea pigs, and primates…"

Form-deprivation myopia was also documented in fish and mice. See, for example, ref [1-9].

4. Line 53. Compensation for optical defocus/lenses was also demonstrated in mice; see for example [1, 7].

5. Line 55-59. Grammar. Please, consider rephrasing. "Application of positive lenses, which cause images to form in front of the retina (myopic defocus), results in slowing the rate of ocular elongation and thickening of the choroid, in order to push the retina toward the image plane (Wallman, Wildsoet et al., 1995). Conversely, application of negative lenses results in an increased rate of elongation and thinning the choroid to pull the retina back toward the image plane."

6. Line 60. Grammar. Did you mean "ocular growth"?

7. Line 60-63. It is well-established that visually induced changes in ocular length are the result of a locally driven "retina-to-choroid-to-scleral chemical signaling cascade" that is initiated by a visual stimulus, followed by chemical changes in the retina and choroid, ultimately resulting in altered extracellular matrix (ECM) remodeling of the scleral shell (Troilo, Smith et al., 2019).

I would not call this signaling cascade "chemical". I would suggest the term "biochemical" or "molecular". Moreover, this cascade leads not only to the scleral ECM remodeling, but also influences neurogenesis in the peripheral retina. See ref [10, 11].

8. Lines 74-75. Inappropriate use of term. "…the [pluripotent] cytokine.." The term "pluripotent" is not appropriate in this context. The term "pluripotent" is used with regard to stem cells.

9. Line 75. Please, consider rephrasing. "…response to chemical treatments…"

10. Lines 78-79. Remove sentence "A preliminary report of our findings was presented previously (Summers and Martinez, IOVS 2020 61: E-abstract 3400)."

11. Lines 82-87 and Figure 1. This section is confusing. While your results show that IL-6 expression is increased in the choroid of eyes recovering from form-deprivation, figure 1 shows that IL-6 expression is reduced in the eyes recovering from form-deprivation. Please, explain. Consider removing panels B and C from figure 1 if this is an artifact.

12. Line 90. Consider changing the sub-heading to "IL-6 is up-regulated in response to myopic optical defocus" or similar.

13. Line 91. "Preliminary results from an Affymetrix microarray experiment…" What do you mean by "preliminary"? Do you have doubts in the results?

14. Lines 91-93 and Figure 2. How microarray data were processed? What normalization method was used? How did you identify differentially expressed genes? It appears that used Student's t-test to identify differential genes. This is not appropriate. Correction for multiple testing, such as Bonferroni, FDR-corrected p-values, or Storey's q-values, should be used.

15. Line 105. "(99 – 1738%)". Please, replace all percentages with fold changes +/- standard deviations throughout the manuscript. Provide exact p-value or q-values.

16. Line 109-126 and Figure 3. I am not sure I agree with the interpretation of results. Your data clearly show that the increase in IL-6 expression is transient – the expression quickly increases and then rapidly drops. This suggests that IL-6 may play a role of the early response gene. It is very difficult to interpret these results though because there are no data on how refractions were changing during this period. I suggest adding refractive error data to this section, and please reflect the fact that the increase in IL-6 expression is transient. This should also be discussed in the Discussion.

17. Lines 136-144 and Figure 4. Your data suggest a light-intensity/wave-length-dependent response, with maximum at 700 lux. Do you have an explanation for this phenomenon? This should be discussed in both Results and Discussion.

18. Figure 5. Please, add refractive error data.

19. Line 185. "Choroidal IL-6 Synthesis Is Transcriptionally Regulated by NO." I don't think you have enough data to say that IL-6 is [transcriptionally] regulated by NO.

20. Line 188. I don't think that existing data support the statement that "nitric oxide synthesis via neuronal nitric oxide synthase (nNOS) is [obligatory] for recovery from form deprivation myopia." Please, consider rephrasing this sentence. NOS signaling is just one of many signaling cascades involved in refractive development.

21. Lines 189-192. "Administration of the non-specific inhibitor nitric oxide synthase inhibitor, Na-nitro-L-arginine methyl ester (L-NAME), or the nNOS inhibitor Nw -propyl-L-arginine, blocks recovery due to inhibition of choroidal thickening and dis-inhibition of scleral proteoglycan synthesis." Please, provide reference(s).

22. Line 215. Grammar. Please, correct. "Therefore, as a second [approach to evaluate NO on choroidal] IL-6 transcription…"

23. Line 222. Bonferroni (two "r") correction is used when correction for multiple comparisons is necessary, such as the case of microarray data. In this case, Bonferroni correction is not necessary. Please, correct and provide exact p-values.

24. Line 257. Replace "Il6" with "IL-6"

25. Figure 8 legend. Grammar. "…eyes following 14 days [of] form deprivation…"

26. Discussion. This section needs to be expanded. Your results should be placed in the context of previous research. Please, consider discussing existing literature on choroidal signaling in myopia. For example, reviews of the choroidal signaling can be found in refs [12, 13].

27. Discussion. This section would also benefit from a review of existing literature on the role interleukins in eye physiology and refractive eye development. For example, two recent studies implicated several interleukins in emmetropization; see refs [14, 15]. The paper by Tkatchenko et al., [14] in fact found that interleukin signaling was activated in the retina of monkeys exposed to positive lenses and not in monkeys exposed to negative lenses, which is very relevant to the current study and suggests that there is a continues cascade between the retina and choroid.

28. Line 279-280. I think it is important to discuss the transient nature of IL-6 upregulation and possible implications of this finding.

29. Lines 286-287. "The identification of a chemical mediator of these visually induced changes in ocular growth has been elusive." Literature suggests that there are multiple mediators and it is very unlikely that IL-6 is THE mediator of visually induced changes in eye growth.

30. Line 287. The use of the term "pleiotropic" is not appropriate in this context. Please, consider replacing.

31. Line 289-290. "For these reasons, choroidally derived IL-6 is an attractive candidate as a potential mediator in the retina-to-sclera signaling cascade."

It is unlikely that IL-6 is THE mediator of visually induced changes in eye growth. The cascade regulating emmetropization is much more complicated that a single gene. While IL-6 is up-regulated in response to positive lenses, it does not exhibit a sign-of-defocus-specific expression pattern. Conversely, several other genes exhibiting sign-of-defocus-dependent expression have been identified.

32. Lines 306-307. "…but was unaffected at 1.5 and 3 hrs of recovery, suggesting that changes in IL-1B gene expression are downstream to that of IL-6." You don't have sufficient experimental data to make that statement, unless you can provide evidence from the literature that there is a link between IL-6 and IL-1B.

33. Line 354. "…that atropine acts [directly] on the choroid to stimulate IL-6 gene expression." You don't have enough experimental evidence to make this statement. Your data clearly suggest that atropine affects IL-6 expression, but it is unknown whether atropine acts directly on the choroid.

34. Methods. Line 487. Typo. Please, correct. "…ratios between [0.08] and 1.9…"

35. Lines 496-497. Grammar. "…each sample [was and] processed with the Affymetrix 3' IVT Express Kit (ThermoFisher Scientific).

36. Methods. Microarray Analyses. Please, describe how microarray data were processed. Describe how gene intensities were extracted, describe the normalization method and how differential genes were identified. Make sure you use appropriate statistical correction for multiple comparisons.

37. Lines 557-558. Please, explain when you used parametric and non-parametric statistics to analyze the data. Provide details about the application of Bonferroni correction.

References

1. Tkatchenko TV, Shen Y, Tkatchenko AV. Mouse experimental myopia has features of primate myopia. Invest Ophthalmol Vis Sci. 2010;51(3):1297-303.

2. Barathi VA, Boopathi VG, Yap EP, Beuerman RW. Two models of experimental myopia in the mouse. Vision Res. 2008;48(7):904-16.

3. Schaeffel F. The mouse as a model for myopia, and optics of its eye. In: Chalupa LM, Williams RW, editors. Eye, retina, and visual system of the mouse. Cambridge, Massachusetts: The MIT Press; 2008. p. 73-85.

4. Faulkner AE, Kim MK, Iuvone PM, Pardue MT. Head-mounted goggles for murine form deprivation myopia. J Neurosci Methods. 2007;161(1):96-100.

5. Schaeffel F, Burkhardt E, Howland HC, Williams RW. Measurement of refractive state and deprivation myopia in two strains of mice. Optom Vis Sci. 2004;81(2):99-110.

6. Tejedor J, de la Villa P. Refractive changes induced by form deprivation in the mouse eye. Investigative Ophthalmology and Visual Science. 2003;44(1):32-6.

7. Jiang X, Kurihara T, Kunimi H, Miyauchi M, Ikeda SI, Mori K, et al., A highly efficient murine model of experimental myopia. Sci Rep. 2018;8(1):2026.

8. Shen W, Sivak JG. Eyes of a lower vertebrate are susceptible to the visual environment. Invest Ophthalmol Vis Sci. 2007;48(10):4829-37.

9. Shen W, Vijayan M, Sivak JG. Inducing form-deprivation myopia in fish. Invest Ophthalmol Vis Sci. 2005;46(5):1797-803.

10. Fischer AJ, Reh TA. Identification of a proliferating marginal zone of retinal progenitors in postnatal chickens. Developmental biology. 2000;220(2):197-210.

11. Tkatchenko AV, Walsh PA, Tkatchenko TV, Gustincich S, Raviola E. Form deprivation modulates retinal neurogenesis in primate experimental myopia. Proc Natl Acad Sci U S A. 2006;103(12):4681-6.

12. Tkatchenko TV, Tkatchenko AV. Pharmacogenomic approach to antimyopia drug development: pathways lead the way. Trends Pharmacol Sci. 2019;40(11):834-53.

13. Troilo D, Smith EL, 3rd, Nickla DL, Ashby R, Tkatchenko AV, Ostrin LA, et al., IMI – Report on Experimental Models of Emmetropization and Myopia. Invest Ophthalmol Vis Sci. 2019;60(3):M31-M88.

14. Tkatchenko TV, Troilo D, Benavente-Perez A, Tkatchenko AV. Gene expression in response to optical defocus of opposite signs reveals bidirectional mechanism of visually guided eye growth. PLoS biology. 2018;16(10):e2006021.

15. Tkatchenko TV, Shah RL, Nagasaki T, Tkatchenko AV. Analysis of genetic networks regulating refractive eye development in collaborative cross progenitor strain mice reveals new genes and pathways underlying human myopia. BMC Med Genomics. 2019;12(1):113.

---

## [Author Response]

Essential revisions:The reviewers were excited about the novel finding connecting in juvenile chicken models, the pro-inflammatory cytokine, interleukin-6 (IL-6) – synthesized and released in the choroid – to the regulation of axial elongation and refraction of the eye.1) Most of the issues revolved around the connection between NO signaling and IL-6. The reviewers would like either new experiments to better substatiate this connection or perhaps consider removing this concept altogether. The reviewers did not come to a consensus on this point but overall, they all agreed that with changes, publication in eLife is warranted. Therefore, please consider both options.

We have elected to keep the data regarding NO signaling and IL-6 in the manuscript, and have added data which provides more support for the idea that choroidally-generated NO can upregulate IL-6 synthesis (see responses to reviewers below).

2) There are significant changes to the writing required. Please see each point of all three reviews. In addition, higher magnification microscopy is required.

We have improved the microscopic images and have added additional images in the Supplementary Materials section. We have also made all suggested corrections to the manuscript.

Reviewer #1 (Recommendations for the authors):It would be extraordinarily helpful for you to create a schematic representation of your concluding hypothesis – that is, how you think IL-6 fits into the signalling system or network that transforms visual information from the retina into regulation of scleral 'growth' and axial elongation.

We have included a diagram summarizing our concluding hypothesis (Figure 10).

Reviewer #2 (Recommendations for the authors):This is a solid experimental study of the role of IL-6 in the control of the response of chicken eyes to myopic defocus in two paradigms – that in which, after the removal of the diffusers, in eyes made myopic through form-deprivation, the rate of axial elongation slows, and in the paradigm in which myopic defocus imposed with positive lenses similarly slows axial elongation.There is, however, a general issue in the terminology used in the presentation of the results that causes confusion. For example, in Figure 1, it is not clear to me what treated eyes are. The authors need to carefully distinguish between untreated but age-matched control eyes, contralateral control eyes, and eyes that have developed myopia and the are baseline for looking at the recovery process.Figure 1 shows IL-6 immunolocalisation in treated and recovering eyes. I am afraid that the magnifications are too low to clearly see localisation to the structures mentioned, or to see any changes. An image of general cell staining would help in for orientation.

We have tried to clarify the confusion regarding Figure 1. We have added the word “contralateral” in the figure legend to clarify that the “control choroid” is in fact the contralateral control for the recovering choroid. As described above, we have also increased the magnification of the images in Figure 1 and included supplemental figures with accompanying H & E staining of adjacent sections to assist in orientation of the images.

Figure 2 shows that IL-6 mRNA expression increases during recovery by over 10-fold. I cannot see any evidence of a corresponding change in the immunohistochemistry in Figure 1. Why?

Although we do see a significant increase in IL-6 protein by ELISA following 6 hrs of recovery (Fig. 3B), we do not observe a significant increase in IL-6 protein concentration in the isolated choroids after 24 hrs of recovery. The immunochemistry was performed on tissue following 24 hrs of recovery, and therefore we do not expect to see a difference in immunolabeling. As we discussion in the Discussion section of the paper ( lines 354 – 357), we suspect that IL-protein is quickly lost from the choroid after its synthesis, either by entering the circulation or adjacent ocular tissues. We have clarified the time point for the immunochemistry experiment in the figure legend for Figure 1.

There are also some excessive statements made in the Introduction that need correction.Line 39: The estimates of the future prevalence of myopia made by Holden et al., are only predictions. They might become "expectations" if there was general acceptance of the underlying methodology, but I believe that the model on which the predictions are based is fundamentally flawed. Unless the authors have carefully considered the methodology, and find it convincing, the should reword this section. This problem can be fixed by changing the word "expected" to "predicted."

We have replaced the word “expected” with “predicted” (line 38).

Line 41-42: The statement that "the cause of human myopia is not understood" cannot be justified. This statement does not reflect the current state of the literature, although it may be that the authors are not familiar with the literature on human myopia. The simplest way to solve this problem is simply to drop the statement. Otherwise, the authors will need to become more familiar with the literature, which suggests that at least two major causes have been identified, namely intensive education, perhaps captured by nearwork, and deprivation of time outdoors. Both are particularly common in East and Southeast Asia where it is well-known that there is an epidemic of myopia.

We agree with this reviewer in that intense education, nearwork and time outdoors are all associated with myopia development in humans. However, the underlying mechanisms by which these environmental conditions cause ocular elongation is not understood. To address this point, we have modified the sentence on lines 40 – 42 to read:

“Although clinical and experimental studies indicate that normal eye growth (emmetropization) is controlled by visual input (Wallman and Winawer, 2004), the molecular mechanisms underlying myopia development in humans is not understood.”

While I do not want to dispute the next statement, that work on animal models has provided valuable insights into the role that the visual environment plays in control of eye growth, it is not clear how the defocus controls studied in this paper are involved in the effects of nearwork, if they in fact are. The situation is clearer with deprivation of time outdoors, because it links to a well-described role of decreased dopamine release in the development of myopia in the animal models. This is important for the current paper because Nickla and colleagues have shown that dopaminergic agents affect choroidal thickness, suggesting a rather obvious set of further experiments.

Our investigations are exploring the molecular mechanisms of visually guided eye growth. The animal models we used in the present study; form deprivation, recovery from form deprivation, and lens induced myopia and hyperopia, are all well characterized visual manipulations that induce pronounced changes in eye growth in a variety of animal models. The association between near work and myopia is anecdotal; to my knowledge there are no prospective studies demonstrating a causal relationship between near work and myopia. Studies demonstrating a role for time outdoors and protection from myopia are very exciting and interesting, and the link with retinal dopamine is substantiated. However, it is still unclear as to how time outdoors is protective for myopia development. Retinal dopamine may be involved, but it is unknown as to how retinal dopamine may regulate scleral remodeling and axial elongation. Nickla et al., 2013, demonstrated that the dopamine agonist, quinpirole, prevented the development of myopia via a nitric oxide-dependent mechanism. In the present study, we observe that choroidal Il-6 is also regulated by a nitric oxide-dependent mechanism. Taken together, we suspect that dopamine and nitric oxide are both involved in the retina-to-scleral chemical cascade and are located upstream of Il-6. We have added these points in the Discussion section of the manuscript (lines 397 – 406).

Line 45: Normal visual input is not restored by removal of the occluder. Rather, a myopically defocused image is imposed. This would still lack much of the high spatial frequency content eliminated by the occlude, but it is clearly sufficient to restore some normality to control of eye growth.The authors raise the potentially interesting possibility that since myopic defocus increases IL-6 expression, that there is over-expression in myopia eyes, presumably even in adult eyes that no longer respond actively to growth signals. They imply that this could contribute to at least some of the long-term consequences of high myopia in particular. The qualification that needs to be added is that it does not seem that correction reduces the pathological consequences of myopia, but given the long-term nature of the processes in humans, and the fact that correction is rarely constant, it remains an interesting suggestion.

We agree with this reviewer’s comment and have replaced the term “normal visual input” with “myopic defocus” to clarify that upon removal of the occluder, the elongated eye will now experience myopic defocus (of the “normal”, unrestricted visual images). (lines 48 – 49)

Reviewer #3 (Recommendations for the authors):Several deficiencies must be addressed before the paper is published:1. Lines 41-42. I would disagree that "the cause of myopia in humans is not understood." Although many environmental factors may influence refractive eye development, the overwhelming evidence suggest that the cause of myopia is excessive exposure to nearwork and hyperopic optical defocus.

We have clarified this point. See response to Reviewer 2, item 4.

2. Lines 43-44. Grammar. "role of the visual environment [in] ocular growth control."

We have replaced “on” with “in”. (line 43)

3. Lines 44-46. "Deprivation of form vision, through the use of visual "occluders" or "goggles" results in accelerated ocular growth and the development of myopia within a matter of days in chicks, tree shrews, guinea pigs, and primates…"Form-deprivation myopia was also documented in fish and mice. See, for example, ref [1-9].

We have added a comment and references regarding form deprivation myopia in fish and mice. Thank you for the references. (lines 45 – 50)

4. Line 53. Compensation for optical defocus/lenses was also demonstrated in mice; see for example [1, 7].

We have added a comment and reference for optical defocus compensation in mice. (line 53)

5. Line 55-59. Grammar. Please, consider rephrasing. "Application of positive lenses, which cause images to form in front of the retina (myopic defocus), results in slowing the rate of ocular elongation and thickening of the choroid, in order to push the retina toward the image plane (Wallman, Wildsoet et al., 1995). Conversely, application of negative lenses results in an increased rate of elongation and thinning the choroid to pull the retina back toward the image plane."

We have rephrased this section to enhance its readability. (lines 55 – 60)

6. Line 60. Grammar. Did you mean "ocular growth"?

Yes. We have replaced “length” with “growth”. (line 61)

7. Line 60-63. It is well-established that visually induced changes in ocular length are the result of a locally driven "retina-to-choroid-to-scleral chemical signaling cascade" that is initiated by a visual stimulus, followed by chemical changes in the retina and choroid, ultimately resulting in altered extracellular matrix (ECM) remodeling of the scleral shell (Troilo, Smith et al., 2019).I would not call this signaling cascade "chemical". I would suggest the term "biochemical" or "molecular". Moreover, this cascade leads not only to the scleral ECM remodeling, but also influences neurogenesis in the peripheral retina. See ref [10, 11].

We have replaced the term “chemical” with “molecular” and “biochemical” in this section. Additionally, we have modified our statement; “…followed by chemical changes in the retina and choroid…” to “…followed by biochemical and structural changes in the retina and choroid…” and have added the Fischer and Tkatchenko references, as suggested. (Lines 62 – 65)

8. Lines 74-75. Inappropriate use of term. "…the [pluripotent] cytokine.." The term "pluripotent" is not appropriate in this context. The term "pluripotent" is used with regard to stem cells.

We have removed the word, “pluripotent”.

9. Line 75. Please, consider rephrasing. "…response to chemical treatments…"

We have replaced “chemical treatments” with “chemical compounds”. (Line 76)

10. Lines 78-79. Remove sentence "A preliminary report of our findings was presented previously (Summers and Martinez, IOVS 2020 61: E-abstract 3400)."

This sentence has been removed.

11. Lines 82-87 and Figure 1. This section is confusing. While your results show that IL-6 expression is increased in the choroid of eyes recovering from form-deprivation, figure 1 shows that IL-6 expression is reduced in the eyes recovering from form-deprivation. Please, explain. Consider removing panels B and C from figure 1 if this is an artifact.

In response to all three reviewers, we have increased the magnification and resolution of images in Figure 1 to better distinguish immunoreactive cells. Additionally, we have included as Supplementary Figure 1 —figure supplement 1, both HandE stained and immunolabelled images from adjacent serial sections (both longitudinal and cross sections) of control choroids in order to compare immunopositive cells with the histoarchitecture of the choroid.

We did not quantify IL-6 protein in immunolabelled tissues. Instead we quantified IL-6 in choroids and choroid culture medium by ELISA, as we felt this method would be more accurate. As mentioned above (Reviewer 2, item #2) and in the Discussion (lines 35 – 357) we suspect that IL-6 is being released from its site of synthesis and either entering the circulation or adjacent ocular tissues shortly after its synthesis. This may explain why we don’t see an obvious difference in IL-6 immunolabelling in control and recovering choroids.

12. Line 90. Consider changing the sub-heading to "IL-6 is up-regulated in response to myopic optical defocus" or similar.

We have reworded the sub-heading to "IL-6 is up-regulated in response to myopic optical defocus". (Line 89)

13. Line 91. "Preliminary results from an Affymetrix microarray experiment…" What do you mean by "preliminary"? Do you have doubts in the results?

We used the term “preliminary” to mean that the microarray results had not been confirmed by other methods. But we see how this might be misunderstood and have removed the word “preliminary”.

14. Lines 91-93 and Figure 2. How microarray data were processed? What normalization method was used? How did you identify differentially expressed genes? It appears that used Student's t-test to identify differential genes. This is not appropriate. Correction for multiple testing, such as Bonferroni, FDR-corrected p-values, or Storey's q-values, should be used.

We have added these details in a new section in the Methods, entitled, “Microarray data analysis” (lines 590-600).

15. Line 105. "(99 – 1738%)". Please, replace all percentages with fold changes +/- standard deviations throughout the manuscript. Provide exact p-value or q-values.

We have replaced all percentages with fold changes +/- standard deviations and provided exact p-values where possible.

16. Line 109-126 and Figure 3. I am not sure I agree with the interpretation of results. Your data clearly show that the increase in IL-6 expression is transient – the expression quickly increases and then rapidly drops. This suggests that IL-6 may play a role of the early response gene. It is very difficult to interpret these results though because there are no data on how refractions were changing during this period. I suggest adding refractive error data to this section, and please reflect the fact that the increase in IL-6 expression is transient. This should also be discussed in the Discussion.

We agree that the increase in IL-6 is transient and suspect that the rapid, transient increase in IL-6 triggers a variety of other choroidal changes. We have added additional discussion on this point (lines 313 – 329). Since we demonstrate significant changes in Il-6 following short period of recovery or lens wear (1.5 hr – 24 hr), we don’t expect to be able to measure any changes in refraction in that brief period of time, with the equipment that we have. To attempt to address this concern, we did include refractions of chick eyes following 10 days of form deprivation (at the time of recovery, Figure 2 —figure supplement 1) as well as the refractions of chick eyes while wearing the +15D and -15D lenses (Figure 5C).

17. Lines 136-144 and Figure 4. Your data suggest a light-intensity/wave-length-dependent response, with maximum at 700 lux. Do you have an explanation for this phenomenon? This should be discussed in both Results and Discussion.

We would refrain from stating that there is a light-intensity /wave-length-dependent response, based on our data. Choroidal IL-6 mRNA concentrations were all similar following exposure to all lighting conditions; the choroidal IL-6 mRNA concentrations of choroids from birds kept in total darkness were significantly lower than that of choroids from all other lighting conditions. We have tried to clarify this in the paper (lines 330 – 353).

18. Figure 5. Please, add refractive error data.

We have added refractive data for age-matched chicks wearing +15D lenses, -15D lenses, and normal untreated chicks (Figure 5C). This data demonstrates that application of +15D and -15D lenses induced myopia and hyperopia, respectively. (Methods, lines 519 – 523)

19. Line 185. "Choroidal IL-6 Synthesis Is Transcriptionally Regulated by NO." I don't think you have enough data to say that IL-6 is [transcriptionally] regulated by NO.

We have rephrased this subheading to read, “Choroidal IL-6 mRNA Expression in Response to Nitric Oxide” (line 175).

20. Line 188. I don't think that existing data support the statement that "nitric oxide synthesis via neuronal nitric oxide synthase (nNOS) is [obligatory] for recovery from form deprivation myopia." Please, consider rephrasing this sentence. NOS signaling is just one of many signaling cascades involved in refractive development.

We agree that there are many signaling cascades involved in mediating ocular changes associated with visually guided eye growth. Nickla et al., 2009 showed that NO and nNOS were necessary for mediating the changes in choroidal thickness and scleral proteoglycan synthesis associated with recovery from form deprivation myopia and compensation for +10D lenses. We have modified our statement by replacing “obligatory” with “necessary”, and also added more detail about the studies by Nickla et al., (2004 and 2009). (Lines 176 – 181)

21. Lines 189-192. "Administration of the non-specific inhibitor nitric oxide synthase inhibitor, Na-nitro-L-arginine methyl ester (L-NAME), or the nNOS inhibitor Nw -propyl-L-arginine, blocks recovery due to inhibition of choroidal thickening and dis-inhibition of scleral proteoglycan synthesis." Please, provide reference(s).

References have been added here. (lines 178 – 181)

22. Line 215. Grammar. Please, correct. "Therefore, as a second [approach to evaluate NO on choroidal] IL-6 transcription…"

We reworded this sentence to read: “ Therefore, we directly tested the effect of an NO donor on IL-6 gene transcription…” (lines 205 – 206)

23. Line 222. Bonferroni (two "r") correction is used when correction for multiple comparisons is necessary, such as the case of microarray data. In this case, Bonferroni correction is not necessary. Please, correct and provide exact p-values.

We recalculated the statistics without the Bonferroni correction and also provided exact p-values throughout the paper.

24. Line 257. Replace "Il6" with "IL-6"

We have replaced “IL6” with “IL-6” in the subheading. (line 263)

25. Figure 8 legend. Grammar. "…eyes following 14 days [of] form deprivation…"

“for” has been replaced with “of”.

26. Discussion. This section needs to be expanded. Your results should be placed in the context of previous research. Please, consider discussing existing literature on choroidal signaling in myopia. For example, reviews of the choroidal signaling can be found in refs [12, 13].

We have expanded the discussion to include previous studies on retinal and choroidal signaling in myopia. (lines 397 – 406, 428 – 438).

27. Discussion. This section would also benefit from a review of existing literature on the role interleukins in eye physiology and refractive eye development. For example, two recent studies implicated several interleukins in emmetropization; see refs [14, 15]. The paper by Tkatchenko et al., [14] in fact found that interleukin signaling was activated in the retina of monkeys exposed to positive lenses and not in monkeys exposed to negative lenses, which is very relevant to the current study and suggests that there is a continues cascade between the retina and choroid.

We have added additional discussion on the connection between inflammation and myopia (lines 358 – 365), as well as on potential regulators of IL-6 that have been implicated in myopia development (lines 428 – 438).

28. Line 279-280. I think it is important to discuss the transient nature of IL-6 upregulation and possible implications of this finding.

We have added a paragraph in the Discussion on the transient nature of IL-6 upregulation in the context of other studies on IL-6 and in visually guided eye growth. (lines 313 – 329)

29. Lines 286-287. "The identification of a chemical mediator of these visually induced changes in ocular growth has been elusive." Literature suggests that there are multiple mediators and it is very unlikely that IL-6 is THE mediator of visually induced changes in eye growth.

We have modified this sentence to read, “The identification of chemical mediators that translate visual signals to scleral extracellular matrix remodeling to effect changes in eye size have been elusive.” (lines 303-305)

30. Line 287. The use of the term "pleiotropic" is not appropriate in this context. Please, consider replacing.

We have replaced the term “pleiotropic” with the word, “multifunctional”. (line 305)

31. Line 289-290. "For these reasons, choroidally derived IL-6 is an attractive candidate as a potential mediator in the retina-to-sclera signaling cascade."It is unlikely that IL-6 is THE mediator of visually induced changes in eye growth. The cascade regulating emmetropization is much more complicated that a single gene. While IL-6 is up-regulated in response to positive lenses, it does not exhibit a sign-of-defocus-specific expression pattern. Conversely, several other genes exhibiting sign-of-defocus-dependent expression have been identified.

Based on this comment, we have modified our statement to read, “… choroidally derived IL-6 could play an important role in the retina-to-sclera signaling cascade.” (lines 307 – 308)

32. Lines 306-307. "…but was unaffected at 1.5 and 3 hrs of recovery, suggesting that changes in IL-1B gene expression are downstream to that of IL-6." You don't have sufficient experimental data to make that statement, unless you can provide evidence from the literature that there is a link between IL-6 and IL-1B.

We have added additional information and references that demonstrate a link between IL-6 and IL-1B. (lines 368 – 372)

33. Line 354. "…that atropine acts [directly] on the choroid to stimulate IL-6 gene expression." You don't have enough experimental evidence to make this statement. Your data clearly suggest that atropine affects IL-6 expression, but it is unknown whether atropine acts directly on the choroid.

We are confused by this comment. Our data show that application of atropine to chick eyes in vivo and to isolated chick choroids in vitro causes a significant increase in choroidal IL-6 mRNA synthesis. We interpret this data to indicate that atropine can act directly on the choroid to stimulate IL-6 transcription (as opposed to acting on the retina or other tissue first). We have also added new data demonstrating that incubation of isolated choroids with atropine stimulates IL-6 protein synthesis (Figure 9C).

34. Methods. Line 487. Typo. Please, correct. "…ratios between [0.08] and 1.9…"

We have corrected the low 260/230 value to read “0.8”. (line 580)

35. Lines 496-497. Grammar. "…each sample [was and] processed with the Affymetrix 3' IVT Express Kit (ThermoFisher Scientific).

We have removed the extraneous word, “and”.

36. Methods. Microarray Analyses. Please, describe how microarray data were processed. Describe how gene intensities were extracted, describe the normalization method and how differential genes were identified. Make sure you use appropriate statistical correction for multiple comparisons.

We have added these details in a new section in the Methods, entitled, “Microarray data analysis” (lines 590-600).

37. Lines 557-558. Please, explain when you used parametric and non-parametric statistics to analyze the data. Provide details about the application of Bonferroni correction.

We have added details on when we used parametric and non-parametric statistics to analyze our data and when the Bonferroni correction was applied. (Lines 657 – 664)